# TaiwanVQA:
# Benchmarking and Enhancing Cultural Understanding in Vision-Language Models

**Hsin-Yi Hsieh**[†] [¶] [1]    **Shang-Wei Liu**[†] [2]    **Chang-Chih Meng**[‡] [3]  **Chien-Hua Chen**[‡] [3]
**Shuo-Yueh Lin**[§4]    **Hung-Ju Lin**[⋆5]    **Hen-Hsen Huang**[¶] [6] [∗]  **I-Chen Wu**[‡] [3] [∗]

[†]National Center for High-performance Computing, NIAR, Taiwan  [§]National Central University, Taiwan
[‡]National Yang Ming Chiao Tung University, Taiwan  [⋆]National Taiwan University, Taiwan
[¶]Institute of Information Science, Academia Sinica, Taiwan

[1] `hsinmosyi@alumni.ncu.edu.tw`  [2] `104352029@nccu.edu.tw`  [3] `{mcc.cs11,chchen.cs12,icwu}@nycu.edu.tw`
[4] `johnnylin@g.ncu.edu.tw`  [5] `r11922147@csie.ntu.edu.tw`  [6] `hhhuang@iis.sinica.edu.tw`

## Abstract

Vision-language models (VLMs) often struggle with culturally specific content — a challenge largely overlooked by existing benchmarks that focus on dominant languages and globalized datasets. We introduce TAIWANVQA, a VQA benchmark designed for Taiwanese culture to evaluate recognition and reasoning in regional contexts. TAIWANVQA contains 2,736 images and 5,472 manually curated questions covering topics such as traditional foods, public signs, festivals, and landmarks. The official benchmark set includes 1,000 images and 2,000 questions for systematic assessment, with the remainder of the data used as training material. Evaluations on state-of-the-art VLMs reveal strong visual recognition but notable weaknesses in cultural reasoning. To address this, we propose a data augmentation strategy that combines human-annotated and synthesized dialogues to enhance cultural understanding. Fine-tuning yields significant gains on TAIWAN-VQA while maintaining stable performance on other multimodal tasks. To further explore the models' cultural understanding, we conducted an open-ended question answering experiment. The results indicate a notable decline in cultural knowledge generation ($\approx$10–20%), suggesting challenges remain. TAIWANVQA offers a scalable framework for building culturally grounded AI models in low-resource cultures, promoting diversity and fairness in multimodal AI. Our dataset and code are publicly available on Hugging Face and GitHub, respectively.

## 1 Introduction

Multimodal vision-language models (VLMs) have achieved remarkable success in integrating visual and textual information, enabling applications such as image captioning and visual question answering [Li et al., 2023, Dai et al., 2023]. Despite these advances, most existing benchmarks emphasize general-domain knowledge and widely spoken languages, often neglecting the challenges posed by culturally specific content and underrepresented languages [Yue et al., 2024a,b, Fu et al., 2024].

Understanding and reasoning about culturally nuanced content is essential for deploying AI systems in real-world settings [Nayak et al., 2024]. Accurately interpreting traditional symbols, local customs, and region-specific artifacts requires models to possess not only visual recognition capabilities but also contextual and cultural knowledge [Hershcovich et al., 2022].

---

[∗]Corresponding authors

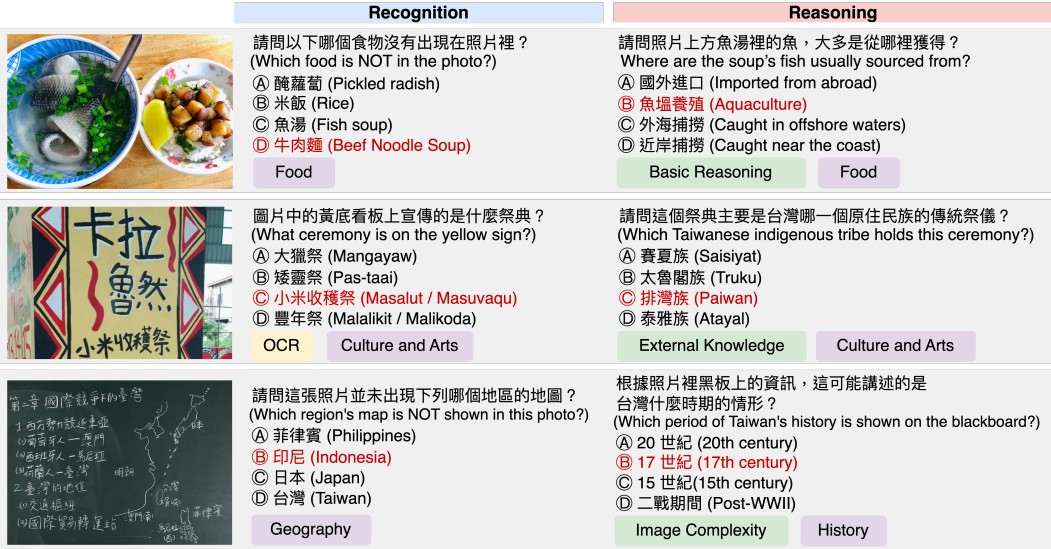

Figure 1: An illustration of the TAIWANVQA benchmark. Each row shows an image paired with two questions: a recognition question (left) and a reasoning question (right), both in multiple-choice format with the correct answers highlighted in red. Below each question, topic categories are labeled in purple (e.g., "attractions", "food"), with additional labels in yellow for OCR requirements in recognition questions and in green for knowledge types in reasoning questions.

However, existing VLM benchmarks are mostly designed around dominant languages and cultural contexts, often overlooking the diversity of localized applications. This limitation poses challenges for evaluating AI systems' ability to adapt to underrepresented cultural settings. Developing benchmarks that incorporate a broader range of cultural perspectives is crucial for ensuring AI models perform reliably across different communities and linguistic backgrounds.

To bridge the gap in evaluating culture-specific multimodal understanding, we introduce a systematic methodology for constructing culturally grounded benchmarks, featuring a structured taxonomy of cultural aspects, annotation guidelines, and design principles applicable across diverse contexts. As an illustrative example, we apply this methodology to develop TAIWANVQA, a visual question answering (VQA) benchmark that evaluates vision-language models (VLMs) on culturally unique content specific to Taiwan (Figure 1).

Built using this methodology, the TAIWANVQA dataset comprises 2,736 images and 5,472 manually curated question–answer pairs, covering topics such as traditional cuisine, festivals, landmarks, and public signage. Among these, 1,000 images and 2,000 QA pairs form the official benchmark set, with the remaining data used for training. We also propose a framework for categorizing cultural knowledge to support reproducibility and adaptability across contexts.

In addition, we explore culture-specific data augmentation using the remaining dataset and show that it improves VLMs' ability to handle localized content. While state-of-the-art VLMs perform well on recognition tasks, they struggle with reasoning that requires deeper cultural understanding. These findings underscore the need for culturally grounded benchmarks like TAIWANVQA to guide the development of more inclusive, context-aware multimodal systems.

Based on these observations, we further conduct an exploratory analysis combining single-choice and open-ended questions, providing methodological insights for assessing vision-language models' cultural understanding capabilities.

As a case study, we establish a scalable and adaptable methodology for building culture-specific multimodal benchmarks, paving the way for the development of more inclusive and culturally aware AI systems across different regions and communities. Our contributions are fivefold:

- **Generalizable Taxonomy of Cultural Knowledge**: We propose a framework for categorizing culture-specific visual questions, distinguishing between recognition and reasoning tasks.

Reasoning tasks are further classified based on the type of external knowledge required, making this taxonomy adaptable to various cultural contexts.

- **Annotation Guidelines for Cultural Data Collection**: We define clear, structured annotation guidelines to ensure consistency, accuracy, and scalability in curating culturally rich datasets, facilitating the creation of benchmarks tailored to different cultural settings.

- **Systematic Framework for Culture-Specific Multimodal Benchmarks**: Using TAIWAN-VQA as a case study, we establish a scalable and adaptable methodology for designing multimodal benchmarks that account for cultural diversity, providing a foundation for extending this approach to other cultural contexts.

- **Comprehensive Evaluation of Vision-Language Models**: We conduct extensive evaluations of state-of-the-art VLMs, including *Gemini2.5* [Comanici et al., 2025] and *GPT-4o* [Hurst et al., 2024], revealing significant gaps in their ability to process and reason about culture-specific content.

- **Culture-Specific Data Augmentation for Model Adaptation**: We introduce a scalable data augmentation strategy that combines human-annotated and synthesized dialogues, demonstrating its effectiveness in enhancing VLMs' ability to understand and reason about culture-specific content. This method provides a practical approach for low-resource cultures to expand training data and develop their own culturally grounded AI models, promoting a more diverse and equitable AI ecosystem.

- **Exploring Cultural Understanding Capabilities:** We conducted exploratory analysis that integrate single-choice and open-ended question formats to rigorously examine the cultural understanding capabilities of vision-language models. This investigation provides methodological insights for assessing models' problem-solving and cultural understanding abilities.

## 2   Related Work

Evaluation of vision–language models (VLMs) has progressed from general visual recognition toward assessing culturally grounded understanding and reasoning. While early benchmarks emphasized image diversity and instance-level recognition, more recent efforts examine whether models can interpret practices, symbols, and knowledge embedded in specific cultural settings.

Early datasets such as DOLLAR STREET [Rojas et al., 2022] and GLDv2 [Weyand et al., 2020] broadened geographic and visual coverage but did not explicitly target cultural knowledge. Cross-lingual/cross-cultural reasoning benchmarks like MARVL [Liu et al., 2021] and multilingual VQA such as MAXM [Changpinyo et al., 2023] advanced evaluation beyond English yet still provide limited coverage of specific local customs and practices. Comprehensive multimodal evaluations (e.g., MMMU [Yue et al., 2024a], MMBENCH [Liu et al., 2024a], SEED-BENCH [Li et al., 2024a], MME [Fu et al., 2024]) assess broad capabilities but are not designed to stress cultural reasoning. In the Chinese context, CMMMU [Zhang et al., 2024a] and CVLUE [Wang et al., 2024] provide valuable multimodal and understanding benchmarks, though they are not culture-specific evaluations.

A growing line of work directly targets cultural knowledge. CULTUREVQA [Nayak et al., 2024] evaluates cultural traditions and artifacts via multiple-choice VQA, and CVQA [Romero et al., 2024] expands to culturally diverse multilingual settings. GLOBALRG [Bhatia et al., 2024] focuses on retrieval and grounding across many countries, while MOSAIC-1.5K [Burda-Lassen et al., 2025] probes cultural captioning. Beyond VQA, CULTURALBENCH [Chiu et al., 2024] assembles a robust, diverse benchmark for cultural knowledge via human–AI teaming at a macro (multi-culture) scope, and CLICK [Kim et al., 2024] offers a culture- and language-focused benchmark centered on the Korean context. Most recently, CULTUREVLM introduces CULTUREVERSE covering over 100 countries [Liu et al., 2025], offering wide geographic breadth through largely web-sourced content. At the domain level, FOODIEQA [Li et al., 2024b] targets fine-grained understanding of Chinese food culture, illustrating how domain-focused benchmarks can isolate specific cultural dimensions (e.g., cuisine) within a broader cultural evaluation landscape.

Building on the above, we present TAIWANVQA, a Traditional-Chinese, Taiwan-centric benchmark that jointly evaluates recognition and cultural reasoning on self-shot images spanning signage, festivals, food, landmarks, and daily practices, with an open license and a transferable collection protocol that reduces pretraining contamination and enables reproducible evaluation.

Table 1: Statistics of recognition and reasoning questions by Type

| Type | Recognition | | | Reasoning | | | |
|------|------|------|------|------|------|------|------|
| | w/ OCR | w/o OCR | All | Basic | External Knowledge | Visual Complex | All |
| **Count** | 344 | 656 | 1,000 | 246 | 674 | 80 | 1,000 |

## 3 Methodology

This section shows the details of our design principles, dataset construction, and evaluation framework, serving as a case study for constructing culture-specific multimodal benchmarks.

### 3.1 Framework for Culture-Specific Benchmarking

Drawing inspiration from recent benchmarks such as MME [Fu et al., 2024] and TRANSPORTA-TIONGAMES [Zhang et al., 2024b], we propose a structured classification of culture-related VQA tasks. This framework categorizes questions into recognition and reasoning-based tasks, with reasoning questions further divided based on the type of external knowledge required.

- **Recognition Questions**: These questions assess a model's ability to identify culturally specific visual elements, such as cuisine, transportation, ecology, and folk activities. The focus is on direct recognition without requiring additional contextual understanding.

- **Reasoning Questions**: These questions evaluate a model's ability to analyze relationships between visual elements (e.g., spatial, contextual, and cultural) while integrating local knowledge to infer meaning. This includes tasks such as understanding symbolic representations, interpreting historical artifacts, and recognizing cultural practices in context.

By incorporating these two question types, our framework enables a comprehensive evaluation of vision-language models' ability to recognize visual features and reason about cultural context. While TAIWANVQA serves as a case study, this methodology is designed to be generalizable to other cultural settings, facilitating the development of more culturally-aware AI systems.

### 3.2 Data Collection

To construct the dataset, we collected 2,736 images featuring Taiwanese culture and daily life, each paired with manually designed recognition and reasoning questions, generating 5,472 questions in total. From these, we selected 1,000 images and their corresponding 2,000 questions as the benchmark data, with the remainder as training materials.

In pursuit of annotation quality and consistency, we recruited nine annotators with diverse backgrounds, varying in residence location, ethnic identity, gender, and academic fields. All annotators underwent a week-long training before participating in image collection and question design. Detailed annotation guidelines are provided in Appendix A.

Beyond the task type classification described in subsection 3.1, we further annotate the benchmark data across three key dimensions to assess models' capabilities in different scenarios:

- **Topic Classification**: Each question is labeled with its corresponding topic. As shown in Figure 2, the dataset primarily focuses on signs and food culture, with the remaining questions evenly distributed across other categories. Detailed topic definitions are provided in Appendix A.2.

- **OCR Requirements**: Recognition questions that require Traditional Chinese text comprehension are specifically marked to evaluate models' optical character recognition (OCR) capabilities.

- **Reasoning Types**: We classify reasoning questions into three categories: Basic Reasoning, where answers can be directly inferred from image content; External Knowledge, which requires an understanding of Taiwanese culture and context beyond visual cues; and Visual Complexity, involving multiple visual elements, spatial relationships, and context-dependent reasoning for accurate interpretation.

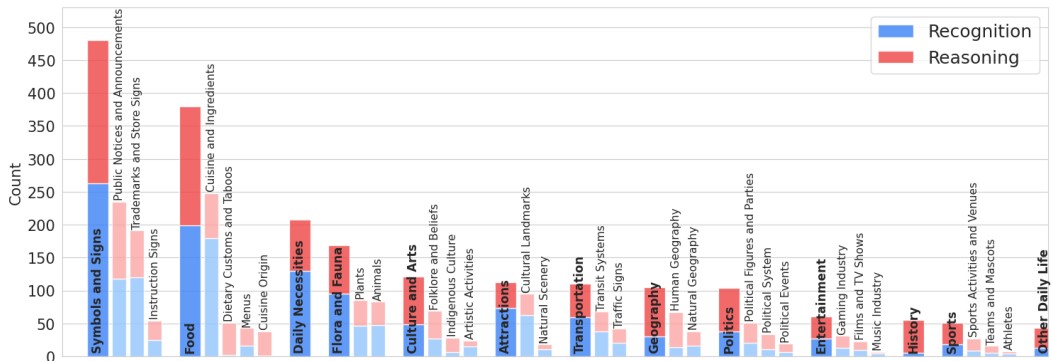

Figure 2: Distribution of question categories. Blue represents recognition questions, red represents reasoning questions. Dark bars show the total number for each topic, light bars show subtopic counts. Absence of light bars indicates topics without subtopics (e.g., Daily Necessities).

Table 2: Quality assessment (Q1-Q8) by four annotators (A1-A4), reported in accuracy (%).

|  | Task Compliance | | | Topic App. | | Content Clarity | | |
| --- | --- | --- | --- | --- | --- | --- | --- | --- |
|  | Q1(n=100) | Q2(n=100) | Q3(n=100) | Q4(n=185) | Q5(n=185) | Q6(n=200) | Q7(n=200) | Q8(n=200) |
| A1 | 94.0 | 98.0 | 95.0 | 97.3 | 97.3 | 100.0 | 99.5 | 99.0 |
| A2 | 93.0 | 99.0 | 97.0 | 97.3 | 94.6 | 100.0 | 100.0 | 100.0 |
| A3 | 100.0 | 91.0 | 94.0 | 93.5 | 91.9 | 100.0 | 100.0 | 99.0 |
| A4 | 99.0 | 96.0 | 98.0 | 90.3 | 99.7 | 98.5 | 98.0 | 96.5 |
| Avg | 96.5 | 96.0 | 96.0 | 94.6 | 95.9 | 99.6 | 99.4 | 98.6 |

Statistics for these annotations are summarized in Table 1. These additional annotations provide a multi-faceted evaluation framework, allowing for a deeper analysis of vision-language models' strengths and limitations in handling culturally specific multimodal content. Through these systematic annotations, we aim to provide a comprehensive evaluation of models' understanding of Taiwanese cultural content.

### 3.3 Data Quality Assessment

TAIWANVQA is manually curated and carefully designed to ensure high-quality cultural representation and question formulation. Despite being created by trained human annotators, the dataset undergoes rigorous evaluation to maintain accuracy, clarity, and adherence to structured guidelines.

To validate its quality, we conducted a systematic evaluation on 10% of randomly sampled data across four key aspects: (1) Question Type Correctness (Q1-2)–ensuring compliance with recognition and reasoning question design guidelines; (2) OCR Compliance (Q3)–verifying the accuracy of OCR requirement labeling in recognition questions; (3) Topic Classification Appropriateness (Q4-5)–ensuring alignment with defined topic and subtopic categories; and (4) Content Clarity (Q6-8)–evaluating question comprehensibility, image clarity, and the necessity of images for answering questions. As shown in Table 2, all criteria achieved over 95% annotator agreement, demonstrating high consistency and reliability in question design and content presentation.

To confirm the necessity of visual information, we extended our evaluation beyond manual inspection of image dependency by comparing the performance of four major VLMs with and without image inputs. Table 3 shows all models performed significantly worse in text-only conditions, reinforcing that TAIWANVQA requires genuine visual reasoning capabilities for accurate responses. This further validates the dataset's role in assessing multimodal understanding in culture-specific contexts.

## 4 Experiments

### 4.1 Experimental Setup

In experiments, we propose a standardized prompt structure ( Figure 3) to ensure consistent evaluation in a zero-shot setting that directly assesses models' intrinsic instruction-following capabilities.

Table 3: Model performance (%) with/without visual information. $\Delta$ indicates the difference computed as (w/o $-$ w/).

| Model | Overall | | | Recognition | | | Reasoning | | |
|---|---|---|---|---|---|---|---|---|---|
| | w/ | w/o | $\Delta$ | w/ | w/o | $\Delta$ | w/ | w/o | $\Delta$ |
| Llama-3.2-90B [Grattafiori et al., 2024] | 51.50 | 11.20 | -40.30 | 62.70 | 7.10 | -55.60 | 40.30 | 15.30 | -25.00 |
| InternVL3-78B [Zhu et al., 2025] | 75.80 | 23.10 | -52.70 | 86.50 | 17.80 | -68.70 | 65.10 | 28.40 | -36.70 |
| Qwen2.5-VL-72B [Bai et al., 2025] | 73.35 | 21.95 | -51.40 | 84.60 | 16.30 | -68.30 | 62.10 | 27.60 | -34.50 |
| Gemini-2.5-pro [Comanici et al., 2025] | 89.35 | 25.75 | -63.60 | 93.40 | 22.20 | -71.20 | 85.30 | 29.30 | -56.00 |
| GPT-4o [Hurst et al., 2024] | 77.40 | 27.70 | -49.70 | 87.30 | 24.60 | -62.70 | 67.50 | 30.80 | -36.70 |

---

**[Question content]**

有以下幾個選項： (Here are the following options:)

  A. \<Option A\>    B. \<Option B\>    C. \<Option C\>    D. \<Option D\>

僅能使用所提供的選項字母（A, B, C, D）作為答案回答，不加任何前綴（例如：答案是）。

(Only use the provided option letters (A, B, C, D) as the answer. Do not add any prefix (e.g., The answer is).)

---

Figure 3: Prompt template for the zero-shot setting.

Table 4: Model performance analysis by different question types (%). Recognition questions are divided into with/without OCR requirements; Reasoning questions include Basic, External Knowledge, and Complex types.

| Model | Overall | Recognition | | | | Reasoning | | | |
|---|---|---|---|---|---|---|---|---|---|
| | | w/ OCR (n=344) | w/o OCR (n=656) | All (n=1000) | | Basic (n=246) | External (n=674) | Complex (n=80) | All (n=1000) |
| Phi-3.5-Vision [Abdin et al., 2024] | 31.05 | 30.81 | 37.50 | 35.20 | | 30.20 | 26.60 | 19.51 | 26.90 |
| Llama-3.2-11B | 32.35 | 45.64 | 45.58 | 45.60 | | 22.86 | 17.83 | 18.29 | 19.10 |
| Llama-3.2-90B | 51.50 | 61.05 | 63.57 | 62.70 | | 45.31 | 40.27 | 25.61 | 40.30 |
| InternVL3-8B | 55.15 | 83.43 | 59.30 | 67.60 | | 48.98 | 42.05 | 29.27 | 42.70 |
| InternVL3-38B | 74.10 | 93.02 | 81.25 | 85.30 | | 72.65 | 59.58 | 60.98 | 62.90 |
| InternVL3-78B | 75.80 | 93.60 | 82.77 | 86.50 | | 74.29 | 62.56 | 58.54 | 65.10 |
| Qwen2.5-VL-7B | 59.75 | 89.53 | 66.01 | 74.10 | | 51.84 | 44.73 | 31.71 | 45.40 |
| Qwen2.5-VL-32B | 65.65 | 92.73 | 69.82 | 77.70 | | 57.55 | 53.79 | 40.24 | 53.60 |
| Qwen2.5-VL-72B | 73.35 | 94.77 | 79.27 | 84.60 | | 67.76 | 60.92 | 54.88 | 62.10 |
| Gemini-2.5-flash | 72.80 | 91.28 | 80.49 | 84.20 | | 68.98 | 60.92 | 42.68 | 61.40 |
| Gemini-2.5-pro | **89.35** | **97.09** | **91.46** | **93.40** | | **89.39** | **84.70** | **78.05** | **85.30** |
| GPT-4o | 77.40 | 91.57 | 85.06 | 87.30 | | 68.57 | 68.50 | 56.10 | 67.50 |

Model predictions are obtained by selecting the option token ("A", "B", "C", or "D") that receives the highest probability among the 20 most probable tokens in the output distribution; if none of these tokens appear, the prediction is counted as incorrect.

Performance is evaluated using accuracy, calculated as $\text{Accuracy} = \frac{N_{\text{correct}}}{N_{\text{total}}}$, where $N_{\text{correct}}$ denotes the number of correctly answered questions and $N_{\text{total}}$ the total number of questions.

Recognizing the sensitivity of language models to the ordering of options Pezeshkpour and Hruschka [2023], we adopt the *CircularEval* strategy [Liu et al., 2024b] (details given in Appendix C), which evaluates responses over four iterations with circular shifts of the answer choices and considers a question correctly answered only if the accurate answer is provided in all iterations.

We evaluate our benchmark using a diverse set of VLMs, including both open-source and proprietary variants. Open-source models comprise leading multilingual and Chinese-based VLMs, while proprietary models include several versions (Detailed in Table 8 and Appendix D). Proprietary models are evaluated via their official APIs, and open-source models are locally deployed using the vLLM framework [Kwon et al., 2023] on DGX-1 V100 GPUs. Our evaluation pipeline is built on lmms-eval,[2] with modifications to meet our experimental requirements; implementation details are provided in Appendix D.

## 4.2 Overall Results

Table 4 presents the performance variations across models in recognition and reasoning tasks. Overall, proprietary models demonstrate a notable competitive advantage over open-source models in

---

[2] https://github.com/EvolvingLMMs-Lab/lmms-eval

understanding Taiwan-specific visual culture, with *Gemini-2.5-pro* and *GPT-4o* achieving the highest accuracy rates, at 89.35% and 77.40%, respectively.

Among open-source models, performance generally scales with model size within the same family. However, a noteworthy observation is that *InternVL3-38B* and *InternVL3-78B* demonstrate comparable competitive performance, suggesting that smaller-scale models are not necessarily at a disadvantage in cultural knowledge comprehension. Additionally, the *InternVL3* and *Qwen2.5-VL* families significantly outperform the *Llama-3.2* series, highlighting the efficiency and effectiveness of these two families in Taiwan-related cultural knowledge.

A consistent trend emerges across all models: recognition tasks yield substantially higher accuracy than reasoning tasks. *Gemini-2.5-pro* leads in both categories, achieving 93.40% in recognition and 85.30% in reasoning, followed closely by *GPT-4o* at 87.30% and 67.50%, respectively.

These results highlight that while state-of-the-art models excel in Taiwan-specific visual recognition, cultural reasoning remains a significant challenge. This underscores not only the need for further improvements in multimodal multicultural reasoning capabilities but also validates the effectiveness of distinguishing task types into *Recognition* and *Reasoning*.

### 4.3   OCR Capability and Cultural Scene Understanding

As part of our data annotation process, we labeled whether each recognition question in our 1,000-question benchmark requires OCR capabilities, allowing us to evaluate models' Traditional Chinese text recognition performance in Taiwanese cultural contexts.

As shown in Table 4, most models perform better on OCR-dependent questions than on non-OCR questions, with the *Phi-3.5-Vision* and *Llama-3.2* series being a notable exception. Not only does this series exhibit lower accuracy on OCR questions compared to non-OCR questions, but its overall scores are also significantly lower than other models, indicating weaker text recognition capabilities.

In contrast, while the *Qwen2.5-VL* series demonstrates strong OCR performance, it experiences significant accuracy drops on non-OCR questions, suggesting that despite their ability to recognize Traditional Chinese text, they struggle with understanding local visual elements such as street scenes, food items, and storefront signs.

Among all models, *Gemini-2.5-pro* maintains consistently high performance across both OCR-dependent and non-OCR tasks, demonstrating not only strong Traditional Chinese text recognition but also a solid understanding of Taiwanese cultural scenes.

This highlights the importance of balancing both text recognition and broader cultural scene comprehension in the development of effective vision-language models.

### 4.4   Types of Reasoning Questions

As shown in Table 4, we analyze the model performance across three types of reasoning questions. Examining the reasoning types, all models perform best in basic reasoning tasks, which is reasonable as these tasks primarily rely on direct visual information inference. Performance slightly decreases in external knowledge tasks that require understanding of Taiwanese cultural context, and models face the greatest challenges in image complexity tasks involving multiple visual elements and spatial relationships.

At this analytical level, *Gemini-2.5-pro* demonstrates an overwhelming advantage over other models. Notably, the *InternVL3* series at medium scale and above, along with *GPT-4o*, also exhibit competitive performance. Overall, this detailed analysis of reasoning types enables us to better understand the performance differences of vision-language models across different cognitive levels, while highlighting the importance of considering diverse reasoning tasks when evaluating model performance.

### 4.5   Model Performance Across Topics

Figure 4 presents model performance across different topic categories, highlighting key trends in recognition and reasoning tasks. *Gemini-2.5-pro* demonstrates consistently strong results across all topics, with *InternVL3-78B*, *GPT-4o*, and *Qwen2.5-VL-72B* following closely as competitive

**MCQ Recognition by Topic**  **MCQ Reasoning by Topic**

Figure 4: Model performance comparison across different topics for recognition and reasoning questions (%).

alternatives. Consistent with previous analyses, *InternVL3-38B* and *InternVL3-78B* exhibit similar performance across many topics, reinforcing the observation that model scale alone may not be the primary factor influencing cultural understanding capabilities.

In recognition tasks, multiple models achieve accuracy exceeding 80% across various topics, such as Signs and Symbols, Sports, and Daily Necessities. Beyond the overall question design of recognition tasks focusing on identifying existing objects in images, this strong performance likely stems from these topics inherently possessing well-defined visual characteristics, making them easier to identify.

However, when tasks require deeper reasoning, even the best-performing models struggle. For example, in the Daily Necessities category, while models accurately recognize objects, their performance drops significantly in reasoning tasks, indicating limitations in contextual understanding beyond simple identification.

A similar trend emerges in topics requiring rich cultural background knowledge, such as Politics, History, and Flora and Fauna, where performance in reasoning tasks remains notably weaker. These findings underscore the persistent challenge VLMs face in bridging the gap between surface-level visual recognition and deep cultural reasoning, particularly in domains that require the integration of diverse cultural content and specialized knowledge.

## 5   Data Augmentation for Culture-Specific VLM Training

While culture-specific benchmarks provide valuable evaluation frameworks, effectively training VLMs to understand and reason about cultural content remains a challenge–particularly when only a limited amount of high-quality labeled data is available. To address this, we propose an automated data augmentation strategy that extends a small manually curated dataset into a large-scale training resource, enabling the development of more culturally aware VLMs. Using TAIWANVQA as a case study, we demonstrate how structured dialogue generation can enhance model adaptation to culture-specific multimodal content, a methodology that can be applied to other underrepresented cultural contexts. We then fine-tune two models on the generated training data and evaluate their performance on both TAIWANVQA and the MMMU[Yue et al., 2024c] benchmark, assessing the impact of this augmentation on culture-specific VLM learning.

### 5.1   Multimodal Data Augmentation

Our training dataset was constructed from 1,736 images held-out from TAIWANVQA. These images are originally paired with 3,472 multiple-choice questions categorized into Recognition and Reasoning types. To make this data suitable for training, we converted all multiple-choice question-answer pairs into structured dialogues, enabling models to learn question-answer relationships in a natural conversational format.

We leveraged automated dialogue generation  [Liu et al., 2023] to expand training coverage. The process consisted of two key steps:

Table 5: Training dataset and model performance comparison.

(a) Expanded training dataset statistics.

| Data Source | Total | Source |
|---|---|---|
| Seed QA pairs | 3,472 | Human |
| Visual Conversation | 1,736 | Generated |
| Attribute Recognition | 1,736 | Generated |
| Contextual Inference | 1,736 | Generated |
| **Total** | **8,680** | **Mixed** |

(b) Performance (%) comparison on TAIWANVQA and MMMU (Zero-Shot) for different models and training conditions.

| | | Llama-3.2-11B | | | Phi-3.5-Vision | | |
|---|---|---|---|---|---|---|---|
| | | base | human | mix | base | human | mix |
| **TaiwanVQA** | Recognition | 45.6 | 51.6 | 61.0 | 35.2 | 36.4 | 38.0 |
| | Reasoning | 19.1 | 27.0 | 36.4 | 26.9 | 28.5 | 28.5 |
| **MMMU** | Valid | 37.7 | 43.7 | 42.8 | 43.2 | 42.8 | 43.1 |
| | Pro-standard | 28.0 | 30.4 | 31.7 | 24.8 | 24.8 | 24.9 |
| | Pro-vision | 5.6 | 11.2 | 13.0 | 8.8 | 11.0 | 11.6 |

1. **Image Captioning**: Using *Qwen2-VL*, we generated detailed captions for each image to serve as the foundation for synthesizing additional dialogues.

2. **Dialogue Generation**: Using *GPT-4o*, we created dialogues encompass three types: Visual Conversation, which broadly discusses the overall visual content; Attribute Recognition, which identifies and explains key attributes of the primary object; and Contextual Inference, which explores the situational context or functional role of the depicted object.

By leveraging *Qwen2-VL*'s strong visual understanding for detailed image captioning and *GPT-4o*'s advanced reasoning and dialogue generation capabilities, our approach ensures precise visual grounding while producing coherent, contextually rich dialogues–resulting in high-quality, culture-specific training data. This augmentation method tripled the number of dialogues per image, expanding the dataset from 3,472 to 8,680 dialogues, as shown in Table 5a.

## 5.2 Fine-Tuning and Evaluation

To assess the impact of additional culture-specific training data, we fine-tuned VLMs using both human-annotated seed data and automatically augmented data. These experiments provide preliminary insights into whether targeted fine-tuning improves model performance on TAIWANVQA while maintaining generalization capabilities. We selected two baseline models as the foundation for our experiments: *Phi-3.5-Vision* and *Llama-3.2-11B*. To systematically evaluate fine-tuning strategies, we defined three model variants:

1. **Base**: Original, non-fine-tuned models.
2. **Human**: Models fine-tuned on human-annotated, seed data.
3. **Mixed**: Models fine-tuned on both seed and augmented training data.

Note that all training materials, including images and questions from both the seed and augmented datasets, are completely separate from the TAIWANVQA benchmark to ensure a fair evaluation. Each baseline model underwent full fine-tuning using the following hyperparameters: a learning rate of $3 \times 10^{-6}$, a batch size of 2, and 2 training epochs. This process generated two fine-tuned versions per model architecture, resulting in four fine-tuned models. Experimental details are in Appendix H.

## 5.3 Performance of Fine-Tuned Models

As shown in Table 5b, both *Llama-3.2-11B* and *Phi-3.5-Vision* exhibited moderate improvements in both recognition and reasoning tasks after being trained on the human or mixed datasets, with Llama demonstrating slightly greater gains on TAIWANVQA. Additionally, we observed a slight increase in accuracy in the pro-vision category of MMMU, which may be attributed to enhanced OCR capabilities, despite the limited scope of the training data. We speculate that the inclusion of synthetic data, combined with the models' inherent multilingual capacity, not only improves domain-specific performance but also helps maintain or even enhance overall comprehension. While these results suggest that fine-tuning for specific domains can enhance performance in culturally relevant tasks, further research is needed to validate the broader applicability of this approach.We consider these pilot results as an initial exploration of how additional training data influences VLM adaptation to localized content.

Table 6: Performance on MCQ and Open-QA(%). Δ indicates the difference computed as (Open-QA − MCQ).

| Model | MCQ | | | Open-QA | | | | | |
|---|---|---|---|---|---|---|---|---|---|
| | All | Recog. | Reason. | All | Δ | Recog. | Δ | Reason. | Δ |
| Phi-3.5-Vision | 31.05 | 35.20 | 26.90 | 10.20 | -20.85 | 12.70 | -22.50 | 7.70 | -19.20 |
| Llama-3.2-11B | 32.35 | 45.60 | 19.10 | 31.60 | -0.75 | 39.00 | -6.60 | 24.20 | +5.10 |
| Llama-3.2-90B | 51.50 | 62.70 | 40.30 | 40.70 | -10.80 | 49.80 | -12.90 | 31.60 | -8.70 |
| InternVL3-8B | 55.15 | 67.60 | 42.70 | 43.55 | -11.60 | 55.30 | -12.30 | 31.80 | -10.90 |
| InternVL3-38B | 74.10 | 85.30 | 62.90 | 51.80 | -22.30 | 65.50 | -19.80 | 38.10 | -24.80 |
| InternVL3-78B | 75.80 | 86.50 | 65.10 | 53.10 | -22.70 | 65.90 | -20.60 | 40.30 | -24.80 |
| Qwen2.5-VL-7B | 59.75 | 74.10 | 45.40 | 50.70 | -9.05 | 64.30 | -9.80 | 37.10 | -8.30 |
| Qwen2.5-VL-32B | 65.65 | 77.70 | 53.60 | 55.85 | -9.80 | 66.80 | -10.90 | 44.90 | -8.70 |
| Qwen2.5-VL-72B | 73.35 | 84.60 | 62.10 | 58.35 | -15.00 | 70.00 | -14.60 | 46.70 | -15.40 |
| Gemini-2.5-flash | 72.80 | 84.20 | 61.40 | 66.80 | -6.00 | 76.40 | -7.80 | 57.20 | -4.20 |
| Gemini-2.5-pro | 89.35 | 93.40 | 85.30 | 71.90 | -17.45 | 79.50 | -13.90 | 64.30 | -21.00 |
| GPT-4o | 77.40 | 87.30 | 67.50 | 67.40 | -10.00 | 77.50 | -9.80 | 57.30 | -10.20 |

# 6 Exploratory Analysis of Open-Ended Question Answering

To examine model performance without reliance on answer-choice cues, we removed multiple-choice (MCQ) options and cast the task as open-ended question answering (Open-QA), aiming to probe internalized, recall-based cultural knowledge rather than option elimination.

Models were prompted with a standardized template that instructs them to provide detailed conclusions while avoiding phrases indicating uncertainty (see Figure 6 in Appendix I). We then used GPT-4.1 as a judge model to evaluate the semantic equivalence of the generated free-form responses against the ground-truth answer. The specific prompts used to guide the judge model are detailed in Appendix I(Figure 7). It is worth emphasizing that, due to the limitations of the evaluation method, the rigor of the results still requires further verification.

As shown in Table 6, the experimental results reveal a significant performance decline across all models in Open-QA, with drops generally ranging from 10-20 percentage points. Notably, GEMINI-2.5-PRO's overall performance decreased from 89.35% to 71.90%, a drop of 17.45 percentage points; INTERNVL3-78B's performance dramatically fell from 75.80% to 53.10%, a substantial decline of 22.70 percentage points. These results highlight the notable limitations of current models in understanding and expressing cultural knowledge.

Compared to MCQ, Open-QA not only requires models to identify the correct answer but also to actively generate and filter relevant knowledge, with evaluation criteria that are more complex and ambiguous. The findings indicate that, without answer choices, current vision-language models still face significant challenges in cultural understanding and expression. Open-QA not only reveals these fundamental difficulties but also provides preliminary validation for the MCQ evaluation method, laying the groundwork for the future development of more refined model assessment frameworks and cross-cultural analysis.

# 7 Conclusion

We introduced TaiwanVQA, a culture-specific VQA benchmark that evaluates VLMs' cultural recognition and reasoning across different knowledge levels. To enhance models' cultural understanding capabilities, we developed a data augmentation strategy combining human-annotated and synthesized dialogues, enabling effective fine-tuning. Additionally, to further validate the models' cultural understanding capabilities, we conducted open-ended question answering experiments. These results highlight the notable limitations of current models in understanding and expressing cultural knowledge. Our work emphasizes the importance of culturally diverse training data and offers a scalable solution for low-resource cultures to build culturally grounded VLMs, contributing to a more diverse, fair, and globally aware AI system.

While our study demonstrates a viable approach, several directions remain for future exploration. Future work could expand evaluation to more Chinese-capable multimodal models, enrich underrepresented domains in the dataset (e.g., Indigenous cultures, religious rituals), validate our methodology on other low-resource cultures, and explore open-ended generative evaluation formats to more comprehensively assess models' cultural understanding.

## Acknowledgments

We would like to acknowledge the support from the National Center for High-performance Computing (NCHC), National Institutes of Applied Research (NIAR), and the National Science and Technology Council (NSTC) under the project "Taiwan's 113th year endeavoring in the promotion of a trustworthy generative AI large language model and the cultivation of literacy capabilities (Trustworthy AI Dialog Engine, TAIDE)".

We also acknowledge the assistance of ChatGPT, Claude and Gemini in refining the writing of this paper. These AI tools were used for editing, restructuring, and enhancing clarity in various sections. All technical content, experimental design, data collection, and analysis were conducted by the authors.

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

## Ethics Statement

All data in TaiwanVQA were exclusively created by our hired annotators, ensuring that there are no copyright concerns or dependencies on third-party proprietary datasets. Both the images and questions were carefully curated and designed to exclude any personal information or identifiable private data. We conducted a second round of annotation review to confirm the absence of privacy risks, adhering strictly to ethical AI and data privacy standards.

We hired nine annotators to design the questions and capture images. Reasonable compensation was provided, and all contributors gave informed consent for their work to be used in research and public release. This study does not involve high-risk human subjects research and does not collect sensitive personal data. According to local regulations, no Institutional Review Board (IRB) approval was required, and all procedures complied with relevant ethical and legal standards in our jurisdiction.

In addition, we have obtained explicit permission to release all materials generated in this project, including images and question-answer pairs, for public use. By releasing this benchmark openly, we aim to support and empower low-resource cultures in building culturally grounded AI systems with significantly reduced effort and cost. Our work promotes greater inclusivity, fairness, and diversity in vision-language model development, encouraging the creation of AI systems that better reflect the richness of global cultures.

## A   Annotation Guidelines

In this section we demonstrate the detail annotation guideline we asked annotator to do. There are three steps in our annotation step. First, we give annotators an general guideline and asked them to take a picture with Taiwan information. Second, we asked annotator to generate a recognition question. Final, we asked annotator to generate a reasoning question.

### A.1   General Guideline

Before the annotators begin annotating data, we first provided them with a general guideline. This guideline asked the annotator follow the rules to write the recognition question and choices, including:

- The primary purpose of data collection: to collect images and questions featuring elements specific to Taiwan.
- Ensuring that the language used in questions reflects common terms and expressions used in Taiwan.
- Ensuring that annotators do not violate any legal issues, such as those related to privacy or copyright.

After reading the overall guideline, the annotator should upload an image containing a Taiwan-specific object.

### A.2   Recognition Question

Next, we asked them to generate a recognition question and corresponding multiple-choice answers. To help annotators understand the guidelines, we provide clear examples and detailed explanations, ensuring both the questions and answer choices meet the required conditions. This guideline introduces key concepts of writing a recognition question, including:

- The definition of a recognition question: questions that assess whether the model can identify and name the object in an image without requiring analysis or inference.
- Emphasize that the question should be answerable solely based on all visible text or clearly identifiable objects in the image, and that the designed options do not include these visible texts or identifiable objects as possible answers.
- Ensure that questions cannot be answered without actually viewing the image.
- If there are multiple objects in the image, specify exactly which person or object to identify to avoid overly simplistic questions.

- Include misleading choices to make it harder for the model to select the correct answer, increasing the challenge.
- No length limit for questions and options.

Additionally, we asked annotators to classify whether the recognition question required ORC capability or not.

Once the question is written, annotators are required to categorize the question's topic. The topics definition is shown in Table 7. This helps in further analyzing the questions and ensuring data quality.

### A.3 Reasoning Question

After writing a recognition question, annotator should write a reasoning question with the guideline. This guideline introduces key concepts of writing a reasoning question, including:

- The definition of a reasoning question: questions that require not only identifying the object but also understanding additional information, such as quantity, use, location, relative position, physical properties, or price, to provide an answer.
- Ensure that questions cannot be answered without actually viewing the image.
- No length limit for questions and options.

Once the reasoning question is written, we also asked the annotator to classify the question topic, similar to the recognition question. Additionally, we asked them to further label the question by identifying the capabilities required to answer it. The annotator should also indicate whether the question requires information about current events.

## B  Topic Definition and Classification Prompt

In this section, we show the detail of the definition of the topics and the analysis of it.

### B.1  Definition

We classify the questions into 13 topics and 27 subtopics. The definition of the topics and subtopics is shown in Table 7.

Table 7: Definition of Each Topic

| Topic | Subtopic | Definition |
|---|---|---|
| Symbols and Signs | | Recognition and understanding of symbols, like priority seating, restrooms, no smoking, etc. |
| | Trademarks and Store Signs | Registered trademarks and store signs, such as FamilyMart, Louisa Coffee, YongChing Real Estate, Hua Nan Bank, etc. |
| | Public Notices and Announcements | Images or text providing information, such as advertisements, banners, usage instructions, and rules. |
| | Instruction Signs | Signs indicating rules or directions, like no smoking, emergency exit, restrooms, priority seating, parking, turn off devices, etc. |
| Attractions | | Including Taiwan's natural and cultural landscapes. |
| | Natural Scenery | Includes Taiwan's mountains, coastlines, lakes, etc., such as Alishan, Taroko National Park, etc. |

| Topic | Subtopic | Definition |
|---|---|---|
| | Cultural Landmarks | Covers Taiwan's historical sites, architectural landmarks, and other non-natural tourist spots, such as Anping Fort in Tainan, Chiang Kai-shek Memorial Hall in Taipei, National Palace Museum, Jiufen Old Street. |
| Food | | Including content related to Taiwan's culinary culture. |
| | Cuisine and Ingredients | Names of dishes and their ingredients, including distinctive foods, components, and garnishes on plates. |
| | Dietary Customs and Taboos | Features of Taiwan's daily dietary habits and customs, including combinations and taboos, like breakfast culture, adding cilantro, etc. |
| | Menus | Judging information based on menu or price list content; images only show text, no actual dishes. |
| | Cuisine Origin | Judging a dish's origin by time or location, or associating it with the culture that originated it. |
| Transportation | | Including content related to Taiwan's transportation. |
| | Transit Systems | Includes Taiwan's metro, train, and bus systems, their operations and features. |
| | Traffic Signs | Covers Taiwan's traffic lights, violation checks, driving tests, etc. |
| Culture and Arts | | Including content related to Taiwan's culture and arts. |
| | Folklore and Beliefs | All things related to culture and religion, including Taiwan's festivals, customs, and taboos like the Mid-Autumn Festival, Dragon Boat Festival, marriage and funeral traditions, religious buildings and decorations, gods, religious practices, temple culture, folk beliefs like Mazu worship. |
| | Indigenous Culture | Taiwan's indigenous customs, languages, and arts, such as those of the Amis and Atayal tribes. |
| | Artistic Activities | Activities like art exhibitions, cultural artifacts, musical instruments, operas, etc. |
| Politics | | Including content related to Taiwan's politics. |
| | Political System | Taiwan's political system and electoral system, such as central and local government bodies, legislative election systems, etc. |
| | Political Events | Activities like elections and social movements. |
| | Political Figures and Parties | Contemporary Taiwanese political figures or parties, such as Lai Ching-te, Chu Li-lun, Taiwan People's Party. |
| Geography | | Including content related to Taiwan's geography. |
| | Natural Geography | Taiwan's landforms and natural features, such as the Central Mountain Range and the eastern coast. |
| | Human Geography | Taiwan's administrative divisions, place name origins, population distribution, industry distribution, etc. |
| Sports | | Including content related to Taiwan's sports and athletics. |
| | Sports | Types of sports and sports venues, such as tennis, badminton, baseball fields. |
| | Athletes | Taiwanese athletes, such as Chuang Chih-yuan, Tai Tzu-ying, Wang Chien-ming. |

| Topic | Subtopic | Definition |
|---|---|---|
| | Teams and Mascots | Taiwan's professional or amateur teams and mascots, such as the Uni Lions, Rakuten Monkeys, Monkeys Kids, Ryan. |
| Flora and Fauna | | Including Taiwan's common flora and fauna. |
| | Animals | Common animal species in Taiwan, such as the Taiwan blue magpie and the Formosan landlocked salmon. |
| | Plants | Common plant species in Taiwan, such as the blackboard tree and large flower impatiens. |
| History | | Covers historical events (e.g., the February 28 Incident, Kaohsiung Incident) and figures who impacted Taiwanese history, such as Chiang Ching-kuo, Lee Teng-hui. |
| Entertainment | | Including content related to Taiwan's entertainment. |
| | Films and TV Shows | Movies, TV series, related events, and venues. |
| | Music Industry | Music genres, important music events, music works, and related venues. |
| | Gaming Industry | Games and industry development. |
| Daily Necessities | | Common items or tools with specific purposes in daily life, requiring identification of the items and their possible uses or purposes. |
| Other Daily Life | | Other content related to the daily lifestyle and habits of Taiwanese people. |

## C  CircularEval strategy

### C.1  Implementation Details

To ensure a robust evaluation of model performance on multiple-choice questions, we implement the CircularEval strategy, as illustrated in Figure 5. This approach mitigates potential biases in model responses caused by the positioning of answer choices.

For instance, consider a question where the model is asked to identify a Taiwanese snack from an image. The original question presents four options: (A) Oyster Omelette, (B) Sweet Potato Balls, (C) Beef Soup, and (D) Oyster Vermicelli, with the correct answer being "Oyster Vermicelli" at position D. CircularEval systematically generates multiple iterations by shifting the answer choices circularly. In the first iteration, the correct answer moves to position C; in the second, it shifts to position B; and in the third, it appears at position A. By evaluating the model's performance across these variations, CircularEval ensures that the model is not biased toward selecting answers based on positional tendencies, thereby providing a more reliable assessment.

## D  Evaluation Experimental Setup

### D.1  Models

We evaluate a diverse set of vision-language models in our experiments, categorized into three groups based on their primary language capabilities and model characteristics.

The first category includes leading **multilingual VLMs**:

**Original Question**

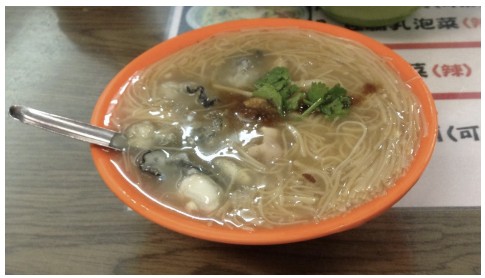

請問照片拍攝的是以下哪種台灣小吃？
*(Which Taiwanese snack is shown in the photo?)*

    A.    蚵仔煎 (Oyster Omelette)
    B.    地瓜球 (Sweet Potato Balls)
    C.    牛肉湯 (Beef Soup)
    D.    蚵仔麵線 (Oyster Vermicelli)

Answer: D

**Four Iterations with Circular Shifts:**

1: A. 蚵仔煎 B. 地瓜球 C. 牛肉湯 D. 蚵仔麵線 → Answer: D
2: A. 地瓜球 B. 牛肉湯 C. 蚵仔麵線 D. 蚵仔煎 → Answer: C
3: A. 牛肉湯 B. 蚵仔麵線 C. 蚵仔煎 D. 地瓜球 → Answer: B
4: A. 蚵仔麵線 B. 蚵仔煎 C. 地瓜球 D. 牛肉湯 → Answer: A

Figure 5: CircularEval example. A model must correctly track the target answer (Oyster Vermicelli) through all shifted positions to be considered successful.

- **Phi-3.5-vision-instruct**: Released by Microsoft under MIT License. See `https://huggingface.co/microsoft/Phi-3.5-vision-instruct`.

- **Llama-based models**: Including Llama-3.2-11B-Vision-Instruct and Llama-3.2-90B-Vision-Instruct. Released by Meta under a custom non-commercial research license. See `https://www.llama.com/models/llama-3/`.

The second category comprises **Chinese-based VLMs**:

- **InternVL3 series**: Consisting of InternVL3-8B-Instruct, InternVL3-38B-Instruct and InternVL3-78B-Instruct. Released by Shanghai AI Laboratory under a custom license. See `https://github.com/OpenGVLab/InternVL`.

- **Qwen2.5-VL series**: Including Qwen2.5-VL-7B-Instruct, Qwen2.5-VL-32B-Instruct and Qwen2.5-VL-72B-Instruct. Released by Alibaba under a non-commercial research license. See `https://github.com/QwenLM/Qwen3-VL`.

The third category consists of **proprietary models**:

- **GPT-4o**: Accessed via the OpenAI API on 2025-10-21, using the model name `chatgpt-4o-latest`. This model is proprietary to OpenAI, with no public release of model weights or architectural details. We accessed the model through the OpenAI API under commercial usage terms, without downloading or distributing any model parameters.

- **Gemini series**: This includes Gemini-2.5-flash and Gemini-2.5-pro, accessed via the Google AI API on 2025-08-06. These models are proprietary to Google and were accessed under standard usage terms. Model weights and specific architectural details are not publicly released.

Table 8: Model specifications of evaluated VLMs. Size is measured in billions of parameters (B).

| Model (Shorthand) | Exact Model Variant | LLM | Vision Encoder | Size |
|---|---|---|---|---|
| Phi-3.5-Vision | Phi-3.5-vision-instruct | Phi-3.5-mini-instruct | CLIP ViT-L/14 | 4.2 |
| Llama-3.2-11B | Llama-3.2-11B-Vision-Instruct | Llama-3.1-8B | ViT-H/14 | 11 |
| Llama-3.2-90B | Llama-3.2-90B-Vision-Instruct | Llama-3.1-70B | ViT-H/14 | 90 |
| InternVL3-8B | InternVL3-8B-Instruct | Qwen2.5-7B | InternViT-300M-448px-V2.5 | 8 |
| InternVL3-38B | InternVL3-38B-Instruct | Qwen2.5-32B | InternViT-6B-448px-V2.5 | 38 |
| InternVL3-78B | InternVL3-78B-Instruct | Qwen2.5-72B | InternViT-6B-448px-V2.5 | 78 |
| Qwen2.5-VL-7B | Qwen2.5-VL-7B-Instruct | Qwen2.5-7B | Redesigned ViT | 7 |
| Qwen2.5-VL-32B | Qwen2.5-VL-32B-Instruct | Qwen2.5-32B | Redesigned ViT | 32 |
| Qwen2.5-VL-72B | Qwen2.5-VL-72B-Instruct | Qwen2.5-72B | Redesigned ViT | 72 |
| Gemini-2.5-flash | – | – | – | – |
| Gemini-2.5-pro | – | – | – | – |
| GPT-4o | – | – | – | – |

Table 9: Chat completion parameters for model inference.

| Parameter | Value | Description |
|---|---|---|
| logprobs | True | Return log probability of output tokens |
| top_logprobs | 20 | Return top 20 likely tokens |
| temperature | 0 | Deterministic sampling |

Table 8 presents the specifications of all evaluated models. For open-source models, we detail their language models, vision encoders, and total parameters in billions (B). The size ranges from 4.2B (Phi-3.5-Vision) to 90B (Llama-3.2-90B) parameters, offering a comprehensive evaluation across different model scales. For proprietary models in the GPT-4o series, these specifications are not publicly available and thus marked with dashes.

## D.2 Implementation Details

In this subsection, we present our experimental configurations for both model inference and deployment. Table 9 shows the chat completion parameters used consistently across all evaluations. For serving open-source models, we utilize the vLLM framework Kwon et al. [2023] to evaluate the performance and scalability of the serving infrastructure under different configurations, which are detailed in Table 10.

The evaluated models include a wide range of vision-language models such as **Qwen2.5-VL**, **InternVL3**, among others. For each model, key configuration parameters were recorded:

- **Maximum Model Length** (`max-model-len`): The maximum sequence length supported by the model.
- **Tensor Parallel Size** (`tensor-parallel-size`): The number of GPUs allocated for parallel inference.
- **GPU Memory Utilization** (`gpu-memory-utilization`): The proportion of GPU memory utilized during serving.
- **Batching Parameters:**
  - **Maximum Number of Batched Tokens** (`num-batched-tokens`): The maximum number of tokens that can be processed in a single batch.
  - **Maximum Number of Sequences** (`max-num-seqs`): The maximum number of sequences processed in parallel.
- **Swap Space** (`swap-space`): Indicates whether disk-based swap space is enabled to handle memory overflow scenarios.

The vLLM framework was used for all experiments. This framework is optimized for high-throughput inference with features such as:

- Token-level pipelining to maximize GPU utilization.

Table 10: Configuration and Status of Vision-Language Models in vLLM Serving Framework. The table summarizes the key parameters used for serving various models, including model length, tensor parallelism, GPU utilization, and batching settings.

| Model | max-model-len | tensor-parallel-size | gpu-memory-utilization | num-batched-tokens | max-num-seqs | swap-space |
|-------|-----------|-------------------|---------------------|-----------------|------------|------------|
| Phi-3.5-Vision | 12888 | 8 | 0.9 | 12888 | 8 | 1 |
| Llama-3.2-11B | 16384 | 8 | 0.85 | 16384 | 8 | 1 |
| Llama-3.2-90B | 16384 | 8 | 0.9 | 16384 | 8 | - |
| InternVL3-8B | 12888 | 4 | 0.85 | 12888 | 8 | 1 |
| InternVL3-38B | 12888 | 8 | 0.85 | 12888 | 8 | 1 |
| InternVL3-78B | 12888 | 8 | 0.85 | 12888 | 4 | 1 |
| Qwen2.5-VL-7B | 16500 | 4 | 0.85 | 16500 | 8 | 1 |
| Qwen2.5-VL-32B | 16384 | 8 | 0.85 | 16384 | 8 | 1 |
| Qwen2.5-VL-72B | 16384 | 8 | 0.95 | 16384 | 4 | 1 |

- Tensor-parallel support for efficient multi-GPU inference.
- Dynamic batching for reducing latency and improving throughput.

Table 10 provides a detailed summary of the experiment configurations and results. These settings can serve as a practical reference for deploying vision-language models in research or production environments.

# E   Experiment Results

Detailed performance results for recognition and reasoning questions across various topics and subtopics are presented in Table 11 and Table 12.

Table 11: Model performance comparison across different topics for recognition and reasoning questions (%). **Topics**: Symbols and Signs (S&S), Attractions (Att), Food, Transportation (Trans), Culture and Arts (C&A), Politics (Pol), Geography (Geo), Sports (Spo), Flora and Fauna (F&F), History (His), Entertainment (Ent), Daily Necessities (DN), and Other Daily Life (ODL).

| Model | S&S | Att | Food | Trans | C&A | Pol | Geo | Spo | F&F | His | Ent | DN | ODL |
|---|---|---|---|---|---|---|---|---|---|---|---|---|---|
| **Recognition Questions** | | | | | | | | | | | | | |
| Phi-3.5-Vision | 35.36 | 29.73 | 32.50 | 39.66 | 46.94 | 23.68 | 30.00 | 38.89 | 26.32 | 20.00 | 25.93 | 49.23 | 30.77 |
| Llama-3.2-11B | 50.95 | 32.43 | 40.00 | 44.83 | 36.73 | 36.84 | 26.67 | 66.67 | 35.79 | 40.00 | 59.26 | 64.62 | 30.77 |
| Llama-3.2-90B | 66.16 | 48.65 | 62.50 | 62.07 | 59.18 | 57.89 | 66.67 | 72.22 | 45.26 | 20.00 | 70.37 | 78.46 | 53.85 |
| InternVL3-8B | 80.61 | 54.05 | 56.50 | 63.79 | 79.59 | 68.42 | 70.00 | 66.67 | 53.68 | 80.00 | 77.78 | 72.31 | 46.15 |
| InternVL3-38B | 91.25 | 77.03 | 84.50 | 81.03 | 87.76 | 76.32 | 93.33 | 94.44 | 76.84 | 100.00 | 77.78 | 86.15 | 92.31 |
| InternVL3-78B | 92.02 | 83.78 | 83.50 | 89.66 | 93.88 | 78.95 | 96.67 | 94.44 | 66.32 | 80.00 | 92.59 | 90.00 | 84.62 |
| Qwen2.5-VL-7B | 85.17 | 68.92 | 68.00 | 68.97 | 75.51 | 73.68 | 90.00 | 83.33 | 54.74 | 80.00 | 81.48 | 73.85 | 69.23 |
| Qwen2.5-VL-32B | 89.35 | 67.57 | 74.00 | 67.24 | 79.59 | 71.05 | 86.67 | 88.89 | 60.00 | 80.00 | 81.48 | 79.23 | 84.62 |
| Qwen2.5-VL-72B | 91.63 | 81.08 | 81.50 | 77.59 | 85.71 | 81.58 | 96.67 | 94.44 | 72.63 | 80.00 | 92.59 | 83.08 | 92.31 |
| Gemini-2.5-flash | 88.21 | 89.19 | 86.00 | 79.31 | 91.84 | 81.58 | 86.67 | 77.78 | 68.42 | 80.00 | 85.19 | 80.77 | 100.00 |
| Gemini-2.5-pro | 96.58 | 95.95 | 96.00 | 86.21 | 91.84 | 89.47 | 96.67 | 94.44 | 82.11 | 100.00 | 96.30 | 92.31 | 100.00 |
| GPT-4o | 88.97 | 91.89 | 89.50 | 86.21 | 91.84 | 81.58 | 90.00 | 94.44 | 71.58 | 80.00 | 88.89 | 89.23 | 76.92 |
| **Reasoning Questions** | | | | | | | | | | | | | |
| Phi-3.5-Vision | 28.57 | 35.90 | 22.10 | 25.49 | 29.17 | 22.73 | 21.33 | 36.36 | 24.32 | 26.00 | 23.53 | 42.31 | 13.33 |
| Llama-3.2-11B | 21.66 | 20.51 | 16.57 | 19.61 | 19.44 | 9.09 | 12.00 | 33.33 | 10.81 | 20.00 | 29.41 | 30.77 | 13.33 |
| Llama-3.2-90B | 46.54 | 33.33 | 36.46 | 39.22 | 40.28 | 33.33 | 26.67 | 51.52 | 33.78 | 46.00 | 41.18 | 57.69 | 26.67 |
| InternVL3-8B | 53.00 | 35.90 | 41.99 | 45.10 | 34.72 | 42.42 | 30.67 | 42.42 | 35.14 | 30.00 | 44.12 | 56.41 | 30.00 |
| InternVL3-38B | 74.65 | 53.85 | 57.46 | 60.78 | 59.72 | 60.61 | 52.00 | 63.64 | 52.70 | 62.00 | 64.71 | 76.92 | 53.33 |
| InternVL3-78B | 72.35 | 64.10 | 61.33 | 66.67 | 63.89 | 66.67 | 58.67 | 69.70 | 52.70 | 62.00 | 64.71 | 73.08 | 60.00 |
| Qwen2.5-VL-7B | 55.76 | 46.15 | 42.54 | 49.02 | 41.67 | 43.94 | 38.67 | 45.45 | 29.73 | 32.00 | 47.06 | 57.69 | 36.67 |
| Qwen2.5-VL-32B | 60.83 | 56.41 | 45.86 | 50.98 | 44.44 | 46.97 | 48.00 | 63.64 | 50.00 | 58.00 | 58.82 | 74.36 | 30.00 |
| Qwen2.5-VL-72B | 66.82 | 53.85 | 60.22 | 64.71 | 62.50 | 66.67 | 61.33 | 66.67 | 48.65 | 64.00 | 64.71 | 73.08 | 30.00 |
| Gemini-2.5-flash | 68.20 | 66.67 | 59.12 | 49.02 | 52.78 | 69.70 | 66.67 | 72.73 | 40.54 | 64.00 | 70.59 | 61.54 | 53.33 |
| Gemini-2.5-pro | 89.86 | 92.31 | 84.53 | 64.71 | 87.50 | 90.91 | 88.00 | 93.94 | 66.22 | 92.00 | 91.18 | 85.90 | 76.67 |
| GPT-4o | 67.28 | 64.10 | 66.85 | 68.63 | 65.28 | 74.24 | 64.00 | 72.73 | 63.51 | 68.00 | 76.47 | 74.36 | 50.00 |

Table 12: Subtopic Performance of Recognition and Reasoning Questions (Accuracy, %). **Abbreviations:** T&S=Trademarks & Store Signs, PN=Public Notices, IS=Instruction Signs, NS=Natural Scenery, CL=Cultural Landmarks, C&I=Cuisine & Ingredients, Men=Menus, CO=Cuisine Origin, TS=Transit Systems, TrS=Traffic Signs, F&B=Folklore & Beliefs, IC=Indigenous Culture, AA=Artistic Activities, PS=Political System, PE=Political Events, PFP=Political Figures & Parties, NG=Natural Geography, HG=Human Geography, SAV=Sports Activities & Venues, Ath=Athletes, T&M=Teams & Mascots, Ani=Animals, Pla=Plants, His=History, FTS=Films & TV Shows, Mus=Music, Gam=Gaming, DN=Daily Necessities, ODL=Other Daily Life.

| | Recognition and Reasoning Performance by Category (Accuracy, %) | | | | | | | | | | | | | | |
| | Recognition | | | | | | | | | | | | | | |
| Model | Symbols & Signs | | | Attractions | | Food | | | Transport | | Culture | | | Geography | |
| | T&S | PN | IS | NS | CL | C&I | Men | CO | TS | TrS | F&B | IC | AA | NG | HG |
| Phi-3.5-Vision | 36.67 | 29.66 | 56.00 | 18.18 | 31.75 | 33.89 | 18.75 | 0.00 | 43.24 | 33.33 | 40.74 | 42.86 | 60.00 | 43.75 | 14.29 |
| Llama-3.2-11B | 65.00 | 34.75 | 60.00 | 45.45 | 30.16 | 42.22 | 25.00 | 0.00 | 56.76 | 23.81 | 33.33 | 0.00 | 60.00 | 25.00 | 28.57 |
| Llama-3.2-90B | 70.00 | 58.47 | 84.00 | 45.45 | 49.21 | 65.56 | 31.25 | 0.00 | 64.86 | 57.14 | 51.85 | 57.14 | 73.33 | 62.50 | 71.43 |
| InternVL3-8B | 87.50 | 75.42 | 72.00 | 63.64 | 52.38 | 58.33 | 37.50 | 0.00 | 67.57 | 57.14 | 77.78 | 85.71 | 80.00 | 68.75 | 71.43 |
| InternVL3-38B | 93.33 | 88.14 | 96.00 | 81.82 | 76.19 | 85.00 | 75.00 | 100.00 | 81.08 | 80.95 | 88.89 | 85.71 | 86.67 | 93.75 | 92.86 |
| InternVL3-78B | 95.83 | 87.29 | 96.00 | 90.91 | 82.54 | 84.44 | 68.75 | 100.00 | 86.49 | 95.24 | 92.59 | 100.00 | 93.33 | 93.75 | 100.00 |
| Qwen2.5-VL-7B | 95.00 | 75.42 | 84.00 | 54.55 | 71.43 | 69.44 | 50.00 | 100.00 | 70.27 | 66.67 | 77.78 | 71.43 | 73.33 | 87.50 | 92.86 |
| Qwen2.5-VL-32B | 95.00 | 84.75 | 84.00 | 63.64 | 68.25 | 74.44 | 62.50 | 100.00 | 75.68 | 52.38 | 85.19 | 57.14 | 80.00 | 75.00 | 100.00 |
| Qwen2.5-VL-72B | 96.67 | 85.59 | 96.00 | 81.82 | 80.95 | 82.22 | 75.00 | 100.00 | 83.78 | 66.67 | 88.89 | 71.43 | 86.67 | 93.75 | 100.00 |
| Gemini-2.5-flash | 93.33 | 83.05 | 88.00 | 90.91 | 88.89 | 87.78 | 62.50 | 100.00 | 86.49 | 66.67 | 96.30 | 85.71 | 86.67 | 75.00 | 100.00 |
| Gemini-2.5-pro | 99.17 | 94.92 | 92.00 | 100.00 | 95.24 | 97.22 | 87.50 | 100.00 | 91.89 | 76.19 | 100.00 | 85.71 | 80.00 | 93.75 | 100.00 |
| GPT-4o | 94.17 | 83.90 | 88.00 | 100.00 | 90.48 | 90.56 | 81.25 | 100.00 | 86.49 | 85.71 | 96.30 | 71.43 | 93.33 | 81.25 | 100.00 |

| | Politics | | | Sports | | | F&F | | His | Entertainment | | | DN | ODL |
| Model | PS | PE | PFP | SAV | Ath | T&M | Ani | Pla | His | FTS | Mus | Gam | DN | ODL |
| Phi-3.5-Vision | 36.36 | 66.67 | 4.76 | 66.67 | 0.00 | 20.00 | 31.25 | 21.28 | 20.00 | 50.00 | 25.00 | 7.69 | 49.62 | 30.77 |
| Llama-3.2-11B | 54.55 | 83.33 | 14.29 | 55.56 | 75.00 | 80.00 | 37.50 | 34.04 | 40.00 | 90.00 | 75.00 | 30.77 | 64.12 | 30.77 |
| Llama-3.2-90B | 81.82 | 83.33 | 38.10 | 77.78 | 50.00 | 80.00 | 43.75 | 46.81 | 20.00 | 90.00 | 50.00 | 61.54 | 78.63 | 53.85 |
| InternVL3-8B | 72.73 | 83.33 | 61.90 | 66.67 | 100.00 | 40.00 | 58.33 | 48.94 | 80.00 | 90.00 | 75.00 | 69.23 | 72.52 | 46.15 |
| InternVL3-38B | 81.82 | 100.00 | 66.67 | 88.89 | 100.00 | 100.00 | 77.08 | 76.60 | 100.00 | 80.00 | 75.00 | 76.92 | 86.26 | 92.31 |
| InternVL3-78B | 72.73 | 100.00 | 76.19 | 88.89 | 100.00 | 100.00 | 62.50 | 70.21 | 80.00 | 100.00 | 100.00 | 84.62 | 90.08 | 84.62 |
| Qwen2.5-VL-7B | 90.91 | 66.67 | 66.67 | 77.78 | 100.00 | 80.00 | 52.08 | 57.45 | 80.00 | 100.00 | 75.00 | 69.23 | 74.05 | 69.23 |
| Qwen2.5-VL-32B | 90.91 | 66.67 | 61.90 | 77.78 | 100.00 | 100.00 | 56.25 | 63.83 | 80.00 | 100.00 | 75.00 | 69.23 | 79.39 | 84.61 |
| Qwen2.5-VL-72B | 81.82 | 66.67 | 85.71 | 88.89 | 100.00 | 100.00 | 70.83 | 74.47 | 80.00 | 100.00 | 75.00 | 92.31 | 83.21 | 92.31 |
| Gemini-2.5-flash | 90.91 | 83.33 | 76.19 | 88.89 | 75.00 | 60.00 | 66.67 | 70.21 | 80.00 | 100.00 | 75.00 | 69.23 | 80.92 | 100.00 |
| Gemini-2.5-pro | 81.82 | 100.00 | 90.48 | 88.89 | 100.00 | 100.00 | 77.08 | 87.23 | 100.00 | 100.00 | 100.00 | 92.31 | 92.37 | 100.00 |
| GPT-4o | 90.91 | 83.33 | 76.19 | 88.89 | 100.00 | 100.00 | 68.75 | 74.47 | 80.00 | 90.00 | 100.00 | 84.62 | 89.31 | 76.92 |

| | Reasoning | | | | | | | | | | | | | | |
| Model | Symbols & Signs | | | Attractions | | Food | | | Transport | | Culture | | | Geography | |
| | T&S | PN | IS | NS | CL | C&I | Men | CO | TS | TrS | F&B | IC | AA | NG | HG |
| Phi-3.5-Vision | 29.58 | 25.64 | 37.93 | 28.57 | 37.50 | 25.00 | 29.63 | 13.51 | 26.67 | 23.81 | 38.10 | 19.05 | 11.11 | 31.82 | 16.98 |
| Llama-3.2-11B | 25.35 | 20.51 | 17.24 | 0.00 | 25.00 | 14.71 | 11.11 | 16.22 | 23.33 | 14.29 | 23.81 | 19.05 | 0.00 | 13.64 | 11.32 |
| Llama-3.2-90B | 53.52 | 36.75 | 68.97 | 0.00 | 40.63 | 39.71 | 14.81 | 37.84 | 43.33 | 33.33 | 47.62 | 23.81 | 44.44 | 45.45 | 18.87 |
| InternVL3-8B | 60.56 | 46.15 | 62.07 | 14.29 | 50.00 | 51.47 | 14.81 | 43.24 | 40.00 | 52.38 | 47.62 | 14.29 | 22.22 | 40.91 | 26.42 |
| InternVL3-38B | 76.06 | 71.79 | 82.76 | 42.86 | 56.25 | 58.82 | 55.56 | 51.35 | 66.67 | 52.38 | 73.81 | 33.33 | 55.56 | 63.64 | 47.17 |
| InternVL3-78B | 76.06 | 69.23 | 75.86 | 42.86 | 68.75 | 67.65 | 59.26 | 56.76 | 70.00 | 61.90 | 71.43 | 42.86 | 77.78 | 68.18 | 54.72 |
| Qwen2.5-VL-7B | 66.20 | 48.72 | 58.62 | 28.57 | 50.00 | 47.06 | 25.93 | 45.95 | 50.00 | 47.62 | 57.14 | 19.05 | 22.22 | 40.91 | 37.74 |
| Qwen2.5-VL-32B | 71.83 | 52.14 | 68.97 | 57.14 | 56.25 | 45.59 | 40.54 | 40.54 | 56.67 | 42.86 | 54.76 | 28.57 | 33.33 | 54.55 | 45.28 |
| Qwen2.5-VL-72B | 73.24 | 61.54 | 72.41 | 57.14 | 53.13 | 61.76 | 55.56 | 59.46 | 66.67 | 61.90 | 71.43 | 52.38 | 44.44 | 68.18 | 58.49 |
| Gemini-2.5-flash | 80.28 | 62.39 | 62.07 | 71.43 | 65.63 | 58.82 | 44.44 | 83.78 | 60.00 | 33.33 | 54.76 | 61.90 | 22.22 | 63.64 | 67.92 |
| Gemini-2.5-pro | 87.32 | 91.45 | 89.66 | 100.00 | 90.63 | 83.82 | 88.89 | 89.19 | 80.00 | 42.86 | 90.48 | 85.71 | 77.78 | 90.91 | 86.79 |
| GPT-4o | 76.06 | 61.54 | 68.97 | 85.71 | 59.38 | 73.53 | 29.63 | 72.97 | 66.67 | 71.43 | 71.43 | 57.14 | 55.56 | 77.27 | 58.49 |

| | Politics | | | Sports | | | F&F | | His | Entertainment | | | DN | ODL |
| Model | PS | PE | PFP | SAV | Ath | T&M | Ani | Pla | His | FTS | Mus | Gam | DN | ODL |
| Phi-3.5-Vision | 21.74 | 53.85 | 10.00 | 55.56 | 0.00 | 18.18 | 22.86 | 25.64 | 26.00 | 15.38 | 33.33 | 27.78 | 42.31 | 13.33 |
| Llama-3.2-11B | 17.39 | 15.38 | 0.00 | 44.44 | 25.00 | 18.18 | 14.29 | 7.69 | 20.00 | 15.38 | 33.33 | 38.89 | 30.77 | 13.33 |
| Llama-3.2-90B | 52.17 | 38.46 | 16.67 | 72.22 | 50.00 | 18.18 | 37.14 | 30.77 | 46.00 | 30.77 | 66.67 | 44.44 | 57.69 | 26.67 |
| InternVL3-8B | 52.17 | 61.54 | 26.67 | 55.56 | 50.00 | 18.18 | 40.00 | 30.77 | 30.00 | 23.08 | 66.67 | 55.56 | 56.41 | 30.00 |
| InternVL3-38B | 73.91 | 69.23 | 46.67 | 88.89 | 50.00 | 27.27 | 65.71 | 41.03 | 62.00 | 53.85 | 66.67 | 72.22 | 76.92 | 53.33 |
| InternVL3-78B | 78.26 | 76.92 | 53.33 | 83.33 | 75.00 | 45.45 | 57.14 | 48.72 | 62.00 | 53.85 | 66.67 | 72.22 | 73.08 | 60.00 |
| Qwen2.5-VL-7B | 60.87 | 53.85 | 26.67 | 50.00 | 75.00 | 27.27 | 37.14 | 23.08 | 32.00 | 38.46 | 66.67 | 50.00 | 57.69 | 36.67 |
| Qwen2.5-VL-32B | 69.57 | 38.46 | 33.33 | 77.78 | 75.00 | 36.36 | 65.71 | 35.90 | 58.00 | 53.85 | 66.67 | 61.11 | 74.36 | 30.00 |
| Qwen2.5-VL-72B | 73.91 | 69.23 | 60.00 | 83.33 | 50.00 | 45.45 | 54.29 | 43.59 | 64.00 | 61.54 | 66.67 | 66.67 | 73.08 | 30.00 |
| Gemini-2.5-flash | 69.57 | 84.62 | 63.33 | 83.33 | 50.00 | 63.64 | 34.29 | 46.15 | 64.00 | 84.62 | 100.00 | 55.56 | 61.54 | 53.33 |
| Gemini-2.5-pro | 95.65 | 92.31 | 86.67 | 94.44 | 100.00 | 90.91 | 65.71 | 66.67 | 92.00 | 92.31 | 100.00 | 88.89 | 85.89 | 76.67 |
| GPT-4o | 82.61 | 92.31 | 60.00 | 83.33 | 50.00 | 63.64 | 65.71 | 61.54 | 68.00 | 69.23 | 66.67 | 83.33 | 74.36 | 50.00 |

Table 13: Conversation Samples Analysis

| Question Type | Sample |
|---|---|
| Recognition | {"from": "human", "value": "請問這是哪一個學位的畢業證書?" (Which degree is this graduation certificate for?) \n<image>"} 

 {"from": "gpt", "value": "農學士" (Bachelor of Agriculture)} |
| Reasoning | {"from": "human", "value": "<image>\n根據畢業證書的內容,請問這所學校即現今台灣的哪所學校?" (According to the graduation certificate, which present-day Taiwanese school does this correspond to?)} 

 {"from": "gpt", "value": "國立台灣大學" (National Taiwan University)} |
| General Visual Conversation | {"from": "human", "value": "這張畢業證書是哪一年的?" (What year is this graduation certificate from?)} 

 {"from": "gpt", "value": "這張畢業證書是1931年的。" (This graduation certificate is from 1931.)} 

 {"from": "human", "value": "證書上寫了什麼內容?" (What does the certificate say?)} 

 {"from": "gpt", "value": "證書上寫著「臺北帝國大學理農學部」,並有「臺北帝國大學理農學部長」的簽名。" (It states 'College of Agriculture, Taipei Imperial University' and is signed by the 'Dean of the College of Agriculture, Taipei Imperial University')} 

 {"from": "human", "value": "證書下方的標牌上有什麼文字?" (What does the plaque read?)} 

 {"from": "gpt", "value": "標牌上寫著「臺北帝大第一屆畢業證書1931年」。" (It reads 'Taipei Imperial University's First Graduation Certificate 1931')} 

 {"from": "human", "value": "這張畢業證書屬於哪所大學?" (Which university is this certificate from?)} 

 {"from": "gpt", "value": "這張畢業證書屬於「臺北帝國大學」。" (This graduation certificate is issued by Taipei Imperial University.)} |
| Attribute Recognition | {"from": "human", "value": "請問這張畢業證書上寫了哪些內容?" (What does the certificate say?)} 

 {"from": "gpt", "value": "證書上寫著「臺北帝國大學理農學部」,並有「臺北帝國大學理農學部長」的簽名。" (It states 'College of Agriculture, Taipei Imperial University' and is signed by the 'Dean of the College of Agriculture, Taipei Imperial University')} 

 {"from": "human", "value": "這張證書與臺北帝國大學有何關聯?" (What is its relation to Taipei Imperial University?)} 

 {"from": "gpt", "value": "這是臺北帝國大學理農學部於1931年頒發的第一屆畢業證書,顯示該校當時的學術發展。" (It is the first graduation certificate awarded by the College of Agriculture in 1931, reflecting the university's academic development at that time.)} |
| Contextual Inference | {"from": "human", "value": "請問這份文件在當時的社會環境中可能代表了什麼樣的意義?" (What might this document represent in its historical context?)} 

 {"from": "gpt", "value": "這是一張「臺北帝國大學理農學部」的畢業證書,標示為「臺北帝大第一屆畢業證書1931年」。在當時的社會環境中,這份證書可能象徵著高等教育的成就,並代表持有者在「臺北帝國大學理農學部」完成學業,具備專業知識與學術資格。" (It is a graduation certificate from the College of Agriculture, Taipei Imperial University, marked as 'Taipei Imperial University's First Graduation Certificate 1931'. In that era, it may symbolize the achievement of higher education and indicate that its holder completed studies at the College, possessing both professional and academic qualifications.)} |

Table 14: Quality Scores for Training Data

| Question Type | Relevance | Coherence | Necessity of Image for Answering | | | |
|---|---|---|---|---|---|---|
| | | | Turn 1 | Turn 2 | Turn 3 | Turn 4 |
| Recognition | 99.75 | 97.50 | 99.50 | - | - | - |
| Reasoning | 100.0 | 98.00 | 99.50 | - | - | - |
| Visual Conv. | 99.25 | 74.00 | 99.50 | 97.00 | 98.75 | 97.75 |
| Attribute Recognition | 99.50 | 96.75 | 98.75 | 53.25 | - | - |
| Contextual Inference | 100.00 | 94.75 | 98.25 | - | - | - |

## F   Training Data Samples

Sample as show in Table 13.

## G   Data Quality Check

Four annotators evaluated the same 100 randomly sampled instances per question type across five types: Recognition, Reasoning, Visual Conversation, Attribute Recognition, and Contextual Inference. For each instance, annotators rated the sample on three criteria:

- **Image-Conversation Relevance**: whether the conversation is related to the image.
- **Conversation Coherence**: whether the conversation flows logically.
- **Necessity of Image for Answering**: for single-turn conversations, this is assessed for the sole question; for multi-turn conversations, each turn is evaluated. In the case of Attribute Recognition, note that while the first turn requires the image to identify the main object, the second turn, asking for an attribute, may sometimes be answerable without the image, leading to a lower score.

For each criterion, annotators provided a binary score (1 if the sample met the criterion, and 0 otherwise), and the quality score for each criterion was computed as the percentage of samples meeting the criterion. The final quality score for each question type is then calculated as the average of the percentages obtained from all four annotators, i.e.,

$$\text{Final Score} = \frac{1}{4} \sum_{i=1}^{4} s_i,$$

where $s_i$ is the percentage score from annotator $i$. The average quality scores (percentage of acceptable instances) for each question type are summarized in Table 14.

## H   Fine-Tuning Experimental Details

We fine-tune the *LLaMA3.2-11B-Vision-Instruct* and *Phi-3.5-Vision-Instruct* models on the Taiwan-VQA training dataset. Input images are uniformly resized to 386x386 pixels, and prompts are formatted using Hugging Face's AutoProcessor with a chat-style template.

Two distinct training sets are employed to generate different model variants:

- **Human:** 3.4k examples manually annotated by human labelers.
- **Mixed:** 8.6k examples in total, combining the 3.4k human-annotated samples with 5.2k synthetic examples.

Each model is trained separately on one of these datasets. All experiments are conducted on 8 NVIDIA H100 GPUs with 512 GB of system memory. The training configuration involves full fine-tuning with a batch size of 2, learning rate of $3 \times 10^{-6}$, BF16 precision, and 3 training epochs. We do not employ parameter-efficient tuning methods such as LoRA.

To mitigate overfitting, token masking is applied to image embedding tokens during training. Apart from image resizing, no additional data augmentation is performed.

Table 15: Performance Analysis of Open-Ended Question Answering (%)

| Model | Overall | Recognition | | | Reasoning | | | |
|---|---|---|---|---|---|---|---|---|
| | | w/ OCR | w/o OCR | All | Basic | External | Complex | All |
| Phi-3.5-Vision | 10.20 | 11.34 | 13.41 | 12.70 | 9.80 | 6.84 | 8.54 | 7.70 |
| Llama-3.2-11B | 31.60 | 38.66 | 39.18 | 39.00 | 30.61 | 21.40 | 28.05 | 24.20 |
| Llama-3.2-90B | 40.70 | 48.26 | 50.61 | 49.80 | 41.63 | 28.68 | 25.61 | 31.60 |
| InternVL3-8B | 43.55 | 71.80 | 46.65 | 55.30 | 43.27 | 27.64 | 31.71 | 31.80 |
| InternVL3-38B | 51.80 | 80.52 | 57.62 | 65.50 | 57.96 | 30.91 | 37.80 | 38.10 |
| InternVL3-78B | 53.10 | 81.40 | 57.77 | 65.90 | 61.22 | 32.39 | 42.68 | 40.30 |
| Qwen2.5-VL-7B | 50.70 | 80.23 | 55.95 | 64.30 | 55.51 | 29.87 | 41.46 | 37.10 |
| Qwen2.5-VL-32B | 55.85 | 83.14 | 58.23 | 66.80 | 61.63 | 37.59 | 54.88 | 44.90 |
| Qwen2.5-VL-72B | 58.35 | 84.01 | 62.65 | 70.00 | 66.12 | 39.23 | 50.00 | 46.70 |
| Gemini-2.5-flash | 66.80 | 86.92 | 70.80 | 76.40 | 71.43 | 52.01 | 57.32 | 57.20 |
| Gemini-2.5-pro | 71.90 | 89.53 | 74.24 | 79.50 | 74.29 | 60.62 | 64.63 | 64.30 |
| GPT-4o | 67.40 | 86.05 | 73.02 | 77.50 | 67.35 | 53.64 | 57.32 | 57.30 |

Training time for the mixed dataset varies depending on model scale:

- *Llama-3.2-Vision-Instruct* takes approximately 1 hour and 30 minutes.
- *Phi-3.5-vision-instruct* completes in about 30 minutes under identical hardware settings.

# I  Open-Ended Question Answering Supplementary Data

## I.1  Detailed Results

Table 15 presents supplementary performance data from the Open QA experiment in the main text's Table 6, including detailed performance of various models across different evaluation dimensions, providing readers with a more comprehensive insight into the experimental findings.

## I.2  Open QA Prompt Templates

For the Open QA evaluation (section 6), we designed distinct prompt templates for the two primary components of the methodology: the model under test and the judge model.

Figure 6 shows the prompt template used for the models under test. This prompt combines an example question (which is presented without multiple-choice options) with a specific instruction to provide a direct conclusion and explicitly avoid phrases indicating uncertainty.

Figure 7 shows the prompt template used to guide the *GPT-4.1* judge model. This prompt provides the necessary details for the evaluation.

---

請問照片拍攝的是以下哪種台灣小吃？
*(Which Taiwanese snack is shown in the photo?)*

請根據圖片以繁體中文提供詳細且具體的答案。
注意：請直接陳述你的結論，嚴禁使用「不確定」、「無法辨識」、「可能是」、「抱歉」或任何迴避性詞語。
*(Please provide a detailed and specific answer in Traditional Chinese based on the image.)*
*(**Note**: Please state your conclusion directly. Evasive words such as "uncertain", "unidentifiable", "maybe", "sorry", or any similar terms are strictly prohibited.)*

---

Figure 6: Prompt template for the model under test in the Open QA setting (section 6). An example question is shown.

You are an expert AI evaluator for a Visual Question Answering task. Your goal is to determine if the candidate's answer is correct.

Question: "[Question content]"

Ground Truth Answer: "[Ground Truth content]"

Candidate's Answer: "[Model Prediction content]"

Please evaluate if the candidate's answer is semantically equivalent to the ground truth. Respond with a valid JSON format containing two keys: "score" (an integer, 1 for correct, 0 for incorrect) and "reasoning" (a brief explanation).

Figure 7: Prompt template for the judge model (section 6).

