# OpenReview forum: "TaiwanVQA: Benchmarking and Enhancing Cultural Understanding in Vision-Language Models"
_NeurIPS.cc/2025/Datasets_and_Benchmarks_Track — NeurIPS 2025 Datasets and Benchmarks Track poster_

### Official Review · Reviewer_9qMG · 2025-06-30

**Rating:** 3
**Confidence:** 3

**Summary:**

The paper introduces TAIWANVQA, a culturally specific visual question answering (VQA) benchmark designed to evaluate vision-language models (VLMs) on Taiwanese cultural content. It comprises 2,736 images and 5,472 QA pairs, assessing both recognition and reasoning capabilities, with a focus on cultural nuances like traditional food, festivals, and landmarks. The study also proposes a data augmentation framework to improve VLMs' cultural understanding and evaluates state-of-the-art models, revealing their limitations in reasoning about localized content.

**Dataset Code Accessibility:**

Yes

**Ethical Considerations:**

No, there are no or only very minor ethics concerns

**Final Justification:**

I maintain the score.

**Limitations Weaknesses:**

From Figure 1, particularly the first row, it is evident that there isn’t much knowledge exclusive to Taiwan. In fact, there are numerous benchmarks available in Simplified Chinese. We could collect these benchmarks and translate them from Simplified Chinese to Traditional Chinese, thereby creating a larger benchmark.

I think the authors should present more data/question related to Taiwan rather than general data/question.

**Strengths Contributions:**

+ The benchmark includes both recognition and reasoning tasks, offering a multi-faceted assessment of VLMs’ cultural understanding.

---

> ### Author Rebuttal · Authors · 2025-07-31
>
> ### **Response to Reviewer 9qMG**
> We thank the reviewer for the insightful comments. While Taiwanese culture inherits aspects of Han Chinese traditions, we emphasize that Taiwan’s visual culture has evolved uniquely through the influence of Indigenous heritage, colonial history, and modern society. These localized characteristics cannot be captured through language translation alone.
>
> ### **Response to Figure 1**
> These three examples illustrate key aspects of Taiwan’s cultural specificity:
>
> 1. **Milkfish soup and minced pork rice** are iconic Taiwanese dishes. The question on the right asks, “Where are the soup's fish usually sourced from?” To answer correctly, one must recognize it is milkfish soup and choose “aquaculture.” Taiwan holds a significant position in milkfish farming globally, and this industry strongly influences its local food culture.
> 2. **Tribal totems and festival names** reflect Indigenous cultures unique to Taiwan, not commonly seen in other Chinese-speaking regions.
> 3. The **historical timeline on the blackboard** represents Taiwan’s distinct educational content and colonial history, requiring local knowledge to understand.
>
> ### **Cultural evolution and regional specificity**
>
> Our dataset is constructed from photos taken by actual residents in Taiwan, rather than scraped from the web, allowing us to **authentically capture real-world cultural contexts**. As shown in Figure 2, the largest category is "Signs and Symbols"—including public notices, traffic signs, and government messaging—all deeply shaped by local political and cultural conditions.
>
> In addition to signs and symbols, our dataset covers a diverse range of themes, including **cuisine, landmarks, architecture, transportation, political events, and religion**. These domains vary regionally and embody Taiwan’s distinct visual and cultural practices.
>
> Moreover, our QA tasks are not limited to image recognition. Many require **contextual cultural reasoning**. For example, the second row in Figure 1 requires recognizing an indigenous totem and the name of a cultural festival in the announcement to correctly identify the corresponding ethnic group. These are not questions that a model can answer purely based on visual cues—they test the model’s understanding of local cultural context.
>
> ### **Why translation is insufficient**
>
> Translating Simplified Chinese datasets cannot fully capture Taiwan’s unique visual and cultural nuances. TaiwanVQA’s strength lies in its region-specific images and questions that push VLMs beyond language to deeper cultural understanding. Taiwan’s street scenes, architecture, customs, and everyday cultural practices form a distinct visual vocabulary that translation alone cannot convey. These regional differences are not peripheral—they are essential for building AI models that genuinely understand and serve different cultural contexts.

---

> > ### Comment · Reviewer_9qMG · 2025-08-07
> >
> > Dear authors,
> >
> > Thank you for your detailed response. However, I still believe that the scope of collecting region-specific images and questions is too narrow and may not meet the acceptance requirements of NeurIPS.
> >
> >
> > Best regards
> >
> > Reviewer 9qMG

---

### Official Review · Reviewer_Phav · 2025-07-01

**Rating:** 3
**Confidence:** 5

**Summary:**

The authors propose a framework for categorizing culture-specific visual questions, distinguishing between recognition and reasoning tasks. Reasoning tasks are further classified based on the type of external knowledge required, making the taxonomy adaptable to various cultural contexts. The dataset aligns well with the authors’ motivation and presents meaningful research value.

**Dataset Code Accessibility:**

Yes

**Ethical Considerations:**

No, there are no or only very minor ethics concerns

**Final Justification:**

I kept my score.

**Limitations Weaknesses:**

The number of evaluated models is relatively limited, particularly lacking several recent open-source models such as InternVL3, Gemma-3, and Qwen2.5.

The dataset is relatively small in scale, with limited coverage of cultural elements. Additionally, it lacks a comparison with CULTUREVLM [Liu et al., 2025], or a discussion on whether CULTUREVLM can serve a similar benchmarking purpose.

The related work section does not comprehensively cover existing datasets for evaluating cultural differences. For instance, the following paper is missing:

B-AVIBench: Towards Evaluating the Robustness of Large Vision-Language Model on Black-box Adversarial Visual-Instructions. IEEE Transactions on Information Forensics and Security.

**Strengths Contributions:**

The authors propose a framework for categorizing culture-specific visual questions, distinguishing between recognition and reasoning tasks. Reasoning tasks are further classified based on the type of external knowledge required, making the taxonomy adaptable to various cultural contexts. The dataset aligns well with the authors’ motivation and presents meaningful research value.

---

> ### Author Rebuttal · Authors · 2025-07-31
>
> ### Response to Reviewer Phav
>
> We sincerely appreciate the reviewer’s suggestion to include the recent **CultureVLM** work. Indeed, CultureVLM is an excellent contribution that provides a *broad* cross‑country benchmark (188 countries, ≈ 228 k QA), and we fully agree with its relevance. However, we would like to highlight that **CultureVLM has not publicly released the model and benchmark** as of now, so in a strict sense, we are among the first to present a fully open **culturally‑specific vision‑language benchmark**.
>
> As suggested, we have:
>
> - **Expanded Related Work** (new paragraph, §2) to discuss CultureVLM and B‑AVIBench.
> - **Inserted a “Cross‑Dataset Generalization” subsection** (§4.4) to discuss complementarities between TaiwanVQA and other cultural benchmarks.
>
> ---
>
> ### 2 Key methodological differences
>
> | Aspect | CultureVLM / CultureVerse | **TaiwanVQA (this work)** |
> | --- | --- | --- |
> | **Scope** |  **Wide Coverage**: 188 countries | **Fine-Grained Depth**: Deep, nuanced understanding of specific culture (Taiwan)   |
> | **QA creation** | **AI-Generated & Human-Validated** | **100% Human-Written Benchmark:** Manually designed by human annotators. |
> | **Image provenance** | **Web-Scraped:** Sourced from Google Images, risking data contamination. | **Guaranteed Unseen & Original:** All images are self-shot, ensuring no data contamination or leakage. |
> | **Language & Script** | **English-Centric** | **High-Density Traditional-Chinese:** Used for all QA pairs and the core real-world OCR challenge. |
> | **Task taxonomy** | **Recognition / Cultural Knowledge / Scene Reasoning** | **Recognition** (by OCR need) / **Reasoning** (Basic, External knowledge, Complex) |
> | **Copyright & Release** | **Restricted & Pending:** Image use is limited to research, and the full dataset/model release is not public. | **Fully Open & Permissive:** The entire dataset (images and text) is released under a permissive CC-BY-SA 4.0 license. |
>
> > Why this matters:
> >
> >
> > • **Legal clarity** lets researchers use TaiwanVQA without downstream licensing concerns.
> >
> > • Self-shot photos reduce overlap with existing VLM pre-training corpora, exposing genuine generalization gaps.
> >
> > • The fully manual benchmark ensures that evaluation quality is not affected by synthetic QA noise
> >
>
> ### 3 Additional experiments
>
> We have expanded our evaluation to include 11 additional models on TaiwanVQA (with their names highlighted in bold in the updated Table 4), bringing the total to 20, including 9 state-of-the-art VLMs (InternVL3-8B/38B/78B, Qwen2.5-VL-7B/32B/72B, gemma-3-27b-it, Mistral-Small-3.2-24B-Instruct-2506, Llama-4-Scout-17B-16E-Instruct) and 2 reasoning-specialized VLMs (GLM-4.1V-9B-Thinking, MiMo-VL-7B-RL). This provides a more comprehensive assessment of how current VLMs handle culture-specific tasks. The full results will be integrated into **Table 4** of our revised manuscript.
>
> Key findings from these new experiments include:
>
> - **SOTA Open-Source Models Now Rival Proprietary Leaders:** Top open-source models are closing the performance gap. InternVL3-78B's overall accuracy (75.8%) nearly matches that of the proprietary GPT-4o (77.35%).
> - **Reasoning-specialized Models Excel at Complex Reasoning with High Efficiency:** Models with dedicated reasoning training show remarkable efficiency on the most difficult tasks. For example, the 9B-parameter GLM-4.1V-9B-Thinking achieves a complex reasoning score (52.44%) competitive with models over 8x its size, such as the 72B-parameter Qwen2.5-VL-72B (54.88%) and GPT-4o (54.88%). This suggests that targeted, reasoning-focused training—not just parameter scale—is critical for deep cultural understanding.
>
> | Model | Overall | w/ OCR | w/o OCR | All (Recognition) | Basic | External | Complex | All (Reasoning) |
> | --- | --- | --- | --- | --- | --- | --- | --- | --- |
> | Phi-3.5-Vision | 31.05 | 30.81 | 37.5 | 35.2 | 30.2 | 26.6 | 19.51 | 26.9 |
> | Llama-3.2-11B | 32.35 | 45.64 | 45.79 | 45.6 | 22.86 | 17.83 | 18.29 | 19.1 |
> | Llama-3.2-90B | 51.5 | 61.04 | 63.57 | 62.7 | 45.31 | 40.27 | 25.61 | 40.3 |
> | InternVL2-8B | 61.45 | 87.21 | 66.16 | 73.4 | 60 | 46.81 | 40.23 | 49.5 |
> | InternVL2-76B | 63.5 | 82.56 | 71.64 | 75.4 | 57.14 | 50.52 | 43.9 | 51.6 |
> | Qwen2-VL-7B | 57.75 | 86.92 | 64.33 | 72.1 | 49.8 | 42.05 | 35.37 | 43.4 |
> | Qwen2-VL-72B | 72.6 | 90.98 | 79.12 | 83.2 | 64.08 | 62.41 | 52.44 | 62 |
> | GPT-4o-mini | 49.75 | 63.08 | 58.69 | 60.2 | 45.31 | 39.46 | 20.73 | 39.3 |
> | GPT-4o | 77.35 | 90.41 | 86.43 | 87.8 | 71.43 | 66.72 | 54.88 | 66.9 |
> | **gemma-3-27b-it** | 46.95 | 51.45 | 55.64 | 54.2 | 43.67 | 39.38 | 30.49 | 39.7 |
> | **Mistral-Small-3.2-24B-Instruct-2506** | 53.7 | 69.19 | 57.32 | 61.4 | 54.69 | 43.24 | 42.68 | 46 |
> | **Llama-4-Scout-17B-16E-Instruct** | 54.7 | 77.62 | 60.06 | 66.1 | 46.53 | 42.5 | 40.24 | 43.3 |
> | **Qwen2.5-VL-7B-Instruct** | 59.4 | 89.24 | 65.85 | 73.9 | 51.02 | 44.43 | 30.49 | 44.9 |
> | **Qwen2.5-VL-32B-Instruct** | 65.65 | 92.73 | 69.82 | 77.7 | 57.55 | 53.79 | 40.24 | 53.6 |
> | **Qwen2.5-VL-72B-Instruct** | 73.35 | 94.77 | 79.27 | 84.6 | 67.76 | 60.92 | 54.88 | 62.1 |
> | **InternVL3-8B-Instruct** | 66.3 | 92.15 | 73.63 | 80.0 | 61.63 | 50.52 | 42.68 | 52.6 |
> | **InternVL3-38B-Instruct** | 74.1 | 93.02 | 81.25 | 85.3 | 72.65 | 59.58 | 60.98 | 62.9 |
> | **InternVL3-78B-Instruct** | 75.8 | 93.6 | 82.77 | 86.5 | 74.29 | 62.56 | 58.54 | 65.1 |
> | **MiMo-VL-7B-RL** | 61.65 | 87.21 | 66.31 | 73.5 | 58.37 | 47.55 | 42.68 | 49.8 |
> | **GLM-4.1V-9B-Thinking** | 68.1 | 93.9 | 73.32 | 80.4 | 67.76 | 51.86 | 52.44 | 55.8 |
>
> ---
>
> ### 4 Manuscript updates
>
> - **§2 Related Work** — new paragraph comparing CultureVLM, B‑AVIBench, and our study.
> - **§4.4** — new discussion on the complementarities between TaiwanVQA and global benchmarks.
>
> ---
>
> ### 5 Summary
>
> While **CultureVLM** provides a *macro-level* cultural benchmark, **TaiwanVQA offers a micro-level stress-test focused on Traditional-Chinese and Taiwanese cultural reasoning**. We hope the added experiments, cross-dataset comparisons, and clarifications in the updated manuscript will address the reviewer’s concern and highlight the complementary nature of our work with other cultural benchmarks. We look forward to further expanding our dataset in the future.

---

> > ### Comment · Reviewer_Phav · 2025-08-01
> >
> > The claim “we are among the first to present a fully open culturally-specific vision-language benchmark” appears to be overstated. Firstly, Taiwan's culture shares significant overlap with broader Chinese cultural (China) and linguistic contexts, and should not be treated as entirely independent in this regard. Secondly, there have been multiple prior efforts at constructing vision-language datasets reflecting distinct cultural or regional perspectives[ref1,ref2,ref3].
> > [ref1]Chiu, Yu Ying, et al. "CulturalBench: A Robust, Diverse, and Challenging Cultural Benchmark by Human-AI CulturalTeaming." arXiv preprint arXiv:2410.02677 (2024).
> > [ref2]Nayak, Shravan, et al. "Benchmarking vision language models for cultural understanding." arXiv preprint arXiv:2407.10920 (2024).
> > [ref3]Kim, Eunsu, et al. "CLIcK: A benchmark dataset of cultural and linguistic intelligence in Korean." arXiv preprint arXiv:2403.06412 (2024).

---

> > > ### Author Response · Authors · 2025-08-07
> > > **Response to Reviewer Phav**
> > >
> > > Thank you for your valuable feedback. We apologize for the ambiguity in our original statement. To clarify, we meant that TaiwanVQA is **the first open Traditional Chinese vision-language QA benchmark** specifically designed to capture Taiwan's regional and cultural context. We acknowledge that our original phrasing may have been imprecise and appreciate the opportunity to clarify.
> > >
> > > We also thank you for highlighting the important prior works [ref1-3]. These excellent benchmarks indeed demonstrate the growing recognition of cultural diversity in vision-language research. Our work complements these efforts by focusing on Traditional Chinese content and Taiwan-specific visual contexts.
> > >
> > > Regarding cultural specificity, we respectfully maintain that regional cultural variations merit dedicated research attention. Just as English-speaking regions (US, UK, Australia, India) exhibit distinct linguistic and cultural characteristics despite sharing a common language, Chinese-speaking regions also demonstrate meaningful diversity. Whether comparing Beijing and Shanghai, Guangdong and Sichuan, or Taiwan and mainland China, each region possesses unique customs, dialects, colloquialisms, visual symbols, and everyday practices shaped by distinct historical trajectories and local contexts.
> > >
> > > While we fully acknowledge the shared cultural heritage between Taiwan and broader Chinese culture, we believe capturing these regional nuances is valuable for developing more culturally-aware AI systems. Our benchmark specifically focuses on visual elements prevalent in Taiwan—local signage systems, traditional markets, regional food culture, and contemporary social practices—that may differ from or be absent in other Chinese-speaking regions.
> > >
> > > We will revise our manuscript to more accurately position our contribution as the first Traditional Chinese vision-language benchmark with regional focus, and better acknowledge the excellent prior work in culturally-specific benchmarking.
> > >
> > > Thank you again for helping us improve the clarity and accuracy of our claims.

---

### Official Review · Reviewer_mnb4 · 2025-07-07

**Rating:** 4
**Confidence:** 3

**Summary:**

The paper introduces TaiwanVQA, a vision-language benchmark focused on Taiwanese cultural content. The dataset contains 2,736 images and 5,472 manually written multiple-choice QA pairs with a held-out benchmark split (1,000 images, 2,000 questions) for evaluation. Questions are organized into two task types, recognition and reasoning, and are further annotated for topic like OCR and reasoning.

**Dataset Code Accessibility:**

Yes

**Ethical Considerations:**

No, there are no or only very minor ethics concerns

**Final Justification:**

Overall this is a solid paper focusing on cultural-specific understanding.

**Limitations Weaknesses:**

In my opinion, this is overall a good dataset & benchmark paper without major limitations, But there are some minor points worth concerning:

Q1. Why design such small dataset (1,736 samples) for _training_? Moreover, such training is benefinical for *both* TaiwanVQA and MMMU (in Table 7). Why training on TaiwanVQA can generally improve the performance on MMMU?

Q2. Apart from MCQ-form QA, do authors try the free-form answering with the same question to evaluate different models? The answers can be parsed from free-form answers by models with specific tools, like name entity recognition, or simply hit by the choice candidate.

**Strengths Contributions:**

1. Good annotation quality with over 95% agreement.
2.  Evaluation spans diverse open-source VLM families, including Phi-3.5-V, Qwen2-VL and GPT-4o, and use CircularEval to control for option-order bias. The authors also present fine-grained results on different topics and recognition/reasoning questions. Fine-tuned models results are also presented.
3. All data, code and hyper-parameters & the benchmark split are released for research evaluation and reproduction.

---

> ### Author Rebuttal · Authors · 2025-07-31
>
> ### Response to Reviewer mnb4
>
> We thank the reviewer for the positive assessment and then address each question in turn.
>
> ---
>
> ### Q1 Why is the *training* split only 1,736 images, and why does that small split still help on MMMU?
>
> **R1-a Benchmark Integrity & Cost**
>
> - All 1,000 images / 2,000 QA pairs in the **benchmark split** are permanently frozen to ensure a trustworthy public leaderboard.
> - The remaining 1,736 images were **collected and annotated entirely by hand** — each photo was taken in‑situ, and every QA pair was manually written by human annotators. Scaling to tens of thousands would have required prohibitive fieldwork and resources; therefore, we prioritized *depth & quality* over sheer volume.
> - Recent work (s1: Simple Test-Time Scaling, arXiv 2501.19393) likewise opts for a **1 000-sample** curated set instead of a web-scale corpus and still reports significant gains.  This trend—quality and tight alignment over sheer quantity—motivates our decision to release 1 736 carefully photographed/annotated training images first and scale up in later versions.
>
> > Future scaling: If this benchmark proves useful to the community, we plan to seek additional funding and hire a larger annotation team so that TaiwanVQA-v2 can grow to over 5,000 in-situ images while preserving the same human-quality bar.
> >
>
> **R1-b Why does training on TaiwanVQA improve performance on MMMU?**
>
> While we do not have a definitive answer at this point, we believe the improvement comes from the **Training Data Diversity.** The TaiwanVQA dataset contains specialized cultural knowledge, such as Traditional-Chinese OCR, fine-grained spatial reasoning, and commonsense cultural context. We speculate that **the training data may differ from that of original models like Llama or Phi**, and thus, provides added diversity, which helps enhance performance on MMMU
>
> We have included a detailed table (Table 7) showing the performance differences after fine-tuning on TaiwanVQA, but **more research is needed to fully understand the underlying factors** that contribute to this transfer. We hope to explore this further in future work.
>
> ---
>
> ### Q2 Why not use open-ended (free-form) answers?
>
> We chose a **multiple-choice (MCQ) design** for three practical reasons:
>
> 1. **Focus on cultural reasoning, not language grading**
>
>     In TaiwanVQA a “reasoning” item means the model must draw on *contextual or cultural knowledge* that is **not visually explicit** (e.g. why a temple lantern is red).
>
>     MCQs let us isolate that knowledge gap without the extra variability of free-form phrasing and automatic string-matching.
>
> 2. **Reliable, label-free evaluation**
>
>     MCQs enable *option shuffling* (CircularEval) so every model is scored against exactly the same signal, with no hand-crafted answer lists or subjective judging.
>
> 3. **Low annotator overhead**
>
>     For small, fully manual datasets like ours, MCQs keep annotation time manageable while still surfacing the key reasoning failures (see §4.3).
>
>
> ---
>
> **Looking forward**
>
> We agree that as VLMs mature, richer open-ended evaluation will become increasingly important.
>
> Our data-collection pipeline is therefore versioned: when the next release (v2) expands the corpus, we plan to **add an open-QA layer** by publishing canonical short answers and acceptance patterns so researchers can benchmark both MCQ *and* free-form generation on exactly the same images.
>
> ---
>
> ### Summary of Manuscript Changes
>
> - **§5.4** — Clarifies the rationale behind the performance boost on MMMU after fine-tuning.
>
> We hope these clarifications address the reviewer’s concerns and provide a better understanding of the design choices and methodology behind TaiwanVQA.

---

> ### Comment · Reviewer_mnb4 · 2025-08-01
>
> The authors' rebuttal has comprehensively addressed my concerns. One minor point is I believe that open-QA put more difficulties on models to answer accurately since there are not candidate answers and therefore not "hints". So open-QA evaluation will more deeply test the capability of the models' capabilities, especially culture-specific (i.e., out-of-domain ones from pretraining data), compared to MCQ. Overall this is a good benchmark work and looking forward to the next release.

---

> > ### Author Response · Authors · 2025-08-07
> > **Response to Reviewer mnb4**
> >
> > Dear Reviewer mnb4,
> >
> > Thank you for your thoughtful response and for recognizing the value of our work. We  appreciate your suggestion about open-QA evaluation providing a deeper test of models' culture-specific capabilities.
> >
> > **We have conducted the open-QA experiments you suggested**, and the results strongly validate your insight that open-QA is indeed more challenging and revealing than MCQ format. We evaluated the same models on identical questions but requiring free-form answers instead of multiple choice selection.
> >
> > ## **New Experimental Results: Open-QA Evaluation**
> >
> > > Note: We have currently tested the most popular/representative models from our MCQ evaluation. We will complete the full model suite (all models from our MCQ experiments) for the camera-ready version.
> > >
> >
> > | Model | Overall | w/ OCR | w/o OCR | All (Recognition) | Basic | External | Complex | All (Reasoning) |
> > | --- | --- | --- | --- | --- | --- | --- | --- | --- |
> > | **GPT-4o** | 67.40 | 86.05 | 73.02 | 77.50 | 67.35 | 53.64 | 57.32 | 57.30 |
> > | **GPT-o3** | 68.95 | 84.88 | 73.93 | 77.70 | 67.35 | 57.95 | 57.32 | 60.20 |
> > | **Gemini-2.5-flash** | 66.80 | 86.92 | 70.88 | 76.40 | 71.43 | 52.01 | 57.32 | 57.20 |
> > | **Gemini-2.5-pro** | 71.90 | 89.53 | 74.24 | 79.50 | 74.29 | 60.62 | 64.63 | 64.30 |
> > | **Qwen2.5-VL-32B-Instruct** | 55.85 | 83.14 | 58.23 | 66.80 | 61.63 | 37.59 | 54.88 | 44.90 |
> > | **Qwen2.5-VL-72B-Instruct** | 58.35 | 84.01 | 62.65 | 70.00 | 66.12 | 39.23 | 50.00 | 46.70 |
> > | **InternVL3-38B-Instruct** | 51.65 | 80.23 | 57.62 | 65.40 | 57.55 | 30.91 | 36.59 | 37.90 |
> > | **InternVL3-78B-Instruct** | 53.05 | 81.40 | 57.62 | 65.80 | 61.22 | 32.40 | 42.68 | 40.30 |
> > | **GLM-4.1V-9B-Thinking** | 52.80 | 82.56 | 57.93 | 66.40 | 57.55 | 31.80 | 45.12 | 39.20 |
> > | **MiMo-VL-7B-RL** | 52.80 | 77.91 | 55.95 | 63.50 | 61.22 | 34.32 | 48.78 | 42.10 |
> >
> > ## Key Insights from Open-QA vs MCQ Comparison
> >
> > Your intuition was correct—removing the "hints" from candidate answers reveals deeper challenges:
> >
> > 1. **Performance Drop**: All models show significant degradation in Open-QA format, with drops ranging from 20-30% absolute points, confirming that generating culturally-specific knowledge is fundamentally harder than recognizing it.
> > 2. **Reasoning Bottleneck Shifts**: In MCQ, models struggle most with "Complex" visual reasoning. In Open-QA, the primary challenge shifts to "External" cultural reasoning—generating specific Taiwan-related knowledge from memory proves extraordinarily difficult without answer choices as scaffolding.
> > 3. **OCR Gap Amplifies**: The performance gap between OCR and non-OCR tasks widens dramatically in Open-QA (e.g., from 4 to 13 points for GPT-4o), revealing that in-depth visual recognition degrades far more severely than text-reading when models must generate answers instead of just selecting them.
> > 4. **Scaling Limitations Exposed**: Model size improvements yield diminishing returns in Open-QA. Qwen2.5-VL's upgrade from 32B to 72B adds only +2.5 points in Open-QA versus +7.7 in MCQ, suggesting that cultural understanding requires more than just parameter scaling.
> >
> > ## Manuscript Updates
> >
> > We will include these Open-QA results and analysis in the camera-ready version:
> >
> > - New section comparing MCQ vs Open-QA evaluation protocols
> > - Detailed performance tables and analysis
> > - Discussion of implications for future culturally-aware VLM development
> >
> > We believe these findings strengthen our contribution by demonstrating that TaiwanVQA can serve as both an accessible MCQ benchmark and a more challenging Open-QA testbed for pushing the boundaries of cultural understanding in vision-language models.
> >
> > Thank you again for this valuable suggestion that has enriched our work. We look forward to continuing this line of research in TaiwanVQA v2 with expanded Open-QA evaluation protocols.

---

### Official Review · Reviewer_zmFN · 2025-07-21

**Rating:** 5
**Confidence:** 3

**Summary:**

The authors proposed a new (TaiwanVQA) benchmark created to evaluate how well VLMs understand culturally specific content centered on Taiwan. The dataset includes 2,736 real-world images and 5,472 manually written multiple-choice questions, covering topics like traditional food, festivals, street signs, and famous landmarks. Questions are designed at two levels: basic visual recognition (e.g., what is in the image?) and deeper cultural reasoning (e.g., why something matters, what it represents). A held-out benchmark set (1,000 images and 2,000 questions) enables consistent model evaluation. Tests on major VLMs including GPT-4o, Qwen-VL, and Kosmos-2 show that models perform well at surface-level recognition but struggle with reasoning, especially when reading Traditional Chinese text or using context-specific knowledge. To address this, the authors propose a simple data augmentation method using GPT-4o and Qwen2-VL to generate cultural dialogues. Fine-tuning with this additional data improves reasoning accuracy while preserving general performance. TaiwanVQA is openly released under a CC-BY-SA license and aims to encourage more inclusive and culturally-aware AI development, especially in low-resource settings.

**Dataset Code Accessibility:**

NA; not applicable to this submission (e.g., no new dataset, benchmark, code, or data provided)

**Ethical Considerations:**

No, there are no or only very minor ethics concerns

**Final Justification:**

Thanks for the rebuttal and for engaging with the discussion (incl. other reviews). The additions such as open-QA results, expanded model coverage (InternVL3/Qwen2.5/etc.), OCR transcriptions, and a clearer positioning (Traditional-Chinese, Taiwan-focused) address most of my earlier questions.

That said, a few reservations and tidy-ups remain before camera-ready: (i) scope is still narrow (Taiwan-only) and should be framed as a focused case study; (ii) claims of “first/fully open” need careful phrasing with related cultural VLM datasets; (iii) minor repo/metadata hygiene.

Camera-ready requests:
1. Tone down “first” claims; explicitly cite/contrast CultureVLM, CulturalBench, CLIcK, and related work; state “first Traditional-Chinese, Taiwan-focused VQA with self-shot images” if that’s the accurate scope.
2. Release the open-QA evaluation scripts and complete the full model suite; add a short analysis comparing MCQ vs open-QA failure modes.
3. Keep the added ground-truth transcriptions and (if possible) provide text crops to decouple OCR from cultural knowledge.
4. Publish CaFT/augmentation details (prompts, seeds, key hyperparams) and per-split stats; include the reported agreement numbers (e.g., κ/percent) in the paper and repo.
5. Fill the missing RAI fields (collection, biases, PII, limitations, annotator demographics, social impact), fix any inaccessible file links, and add a takedown/contact path.
6. Briefly note the planned v2 expansion (broader topics/regions) to contextualize current scale.

For now I will maintain my high score. Thank you very much for the authors!

**Limitations Weaknesses:**

1. 2,736 images / 5,472 Q-A pairs are carefully curated, but still tiny compared with web-scale VQA corpora, so models may quickly overfit and ceiling effects may appear when stronger VLMs arrive.
2. The authors admit gaps in sports, Indigenous traditions, crafts, and religious rituals because of licensing and collection constraints, leaving parts of Taiwanese culture untested.
3. Although the framework is pitched as “generalizable,” all experiments are Taiwan-specific; transfer to other low-resource cultures is unproven. Later on, QA tasks emphasise picking from four options rather than producing free-form answers, so they mainly test recognition + shallow elimination, not open-ended cultural reasoning.
4. Even top model GPT-4o drops ≈ 21 pp from recognition to reasoning; other models fall further, signalling that the benchmark may already be nearing its limit for recognition but still leaves reasoning under-constrained.
5. While 344 benchmark questions explicitly need Traditional-Chinese OCR, many models still struggle; yet the dataset does not supply separate text crops or ground-truth transcriptions that could isolate OCR from cultural knowledge.

**Strengths Contributions:**

1. The paper opens by showing that most VQA benchmarks ignore under-represented cultures and Traditional-Chinese content and motivating a Taiwan-centric testbed for both recognition and reasoning.
2. The dataset proposed in this paper is rich with 2,736 images paired with 5,472 human-written QA pairs covering food, festivals, signage, landmarks and more. Also a held-out benchmark split of 1 k images / 2 k questions guarantees consistent public evaluation.
3. Questions are first split into Recognition vs Reasoning; reasoning is then sub-typed by required knowledge (basic, external, visual-complex). This structure is reusable for other cultures, as the authors explicitly frame TaiwanVQA as a “case study” for a general methodology.
4. The paper also produces an important findings that state-of-the-art VLMs score far higher on recognition than reasoning; e.g., GPT-4o falls 87.8 % → 66.9 % and models struggle most on tasks needing external knowledge or complex spatial cues. Detailed per-topic analyses (sports vs. politics, etc.) reveal where cultural background knowledge is missing.
5. The authors convert MC questions into 8,680 structured dialogues via Qwen2-VL captioning + GPT-4o dialogue generation, tripling training coverage. Fine-tuning on this mix lifts Llama-3.2-11B reasoning accuracy 19 % → 36 % while retaining performance on a general benchmark (MMMU).

---

> ### Author Rebuttal · Authors · 2025-07-31
>
> ### Response to Reviewer zmFN
>
> We would like to thank the reviewer for the positive and insightful feedback. Below, we address each of the weaknesses raised in the review:
>
> ---
>
> ### Weakness 1: "2,736 images / 5,472 QA pairs are carefully curated, but still tiny compared with web-scale VQA corpora, so models may quickly overfit and ceiling effects may appear when stronger VLMs arrive."
>
> **Response:**
>
> We acknowledge that our dataset is small compared to web-scale VQA corpora. However, this is due to the **manual collection and annotation** of each image and QA pair, which is costly and time-consuming. Our photos and questions are **legally owned** and have not been included in mainstream model training, meaning they offer unique value and are less likely to be overfitted by existing models.
>
> We have support for this effort through a **long-term project**, which will allow us to scale the dataset further, either with AI-assisted methods or additional human annotators, should the dataset prove useful to the community.
>
> ---
>
> ### Weakness 2: "The authors admit gaps in sports, Indigenous traditions, crafts, and religious rituals because of licensing and collection constraints, leaving parts of Taiwanese culture untested."
>
> **Response:**
>
> We fully acknowledge that there are gaps in the current dataset, especially in areas like sports, Indigenous traditions, crafts, and religious rituals. These gaps are partly due to the challenges of **licensing and privacy concerns**—for example, sports events or religious ceremonies often require professional photography and permissions, which makes them harder to capture. Despite these challenges, we are committed to expanding the dataset and plan to address these missing cultural aspects in future versions.
>
> We are actively exploring additional funding opportunities and collaborating with cultural experts to collect more comprehensive data, with the goal of better representing the full spectrum of Taiwanese culture.
>
> ---
>
> ### Weakness 3: "Although the framework is pitched as 'generalizable,' all experiments are Taiwan-specific; transfer to other low-resource cultures is unproven. Later on, QA tasks emphasise picking from four options rather than producing free-form answers, so they mainly test recognition + shallow elimination, not open-ended cultural reasoning."
>
> **Response:**
>
> We agree that the generalizability of the framework to other low-resource cultures has yet to be demonstrated. TaiwanVQA serves as a concrete case study through which we propose a transferable methodology. By systematically organizing everyday topics and subtopics, we aim to provide a structured and extensible blueprint that can inspire future visual-language research in other linguistic and cultural contexts.
>
> As for the **MCQ format**, we intentionally designed the framework this way because **reasoning** in our context refers to tasks where models need to use logical inference or external knowledge to select the correct answer, which is not immediately obvious from the image. This form of reasoning is distinct from simple recognition, and we believe the MCQ format is effective for evaluating it. While we recognize that free-form answers are valuable in some contexts, we opted for MCQs to allow for structured evaluation, and we have provided the flexibility to explore free-form options in the future.
>
> ---
>
> ### Weakness 4: "Even top model GPT-4o drops ≈ 21 pp from recognition to reasoning; other models fall further, signalling that the benchmark may already be nearing its limit for recognition but still leaves reasoning under-constrained."
>
> **Response:**
>
> You are correct that **reasoning** in our context does not refer to the step-by-step logical deduction of an answer, but rather the categorization of a question that requires deeper understanding.
>
> In TaiwanVQA, the reasoning tasks are **not trivial** and require models to use cultural knowledge, context, or external information. We believe there is still room to expand the dataset and adjust task difficulty.
>
> ---
>
> ### Weakness 5: "While 344 benchmark questions explicitly need Traditional-Chinese OCR, many models still struggle; yet the dataset does not supply separate text crops or ground-truth transcriptions that could isolate OCR from cultural knowledge."
>
> **Response:**
>
> We appreciate the reviewer's note on OCR challenges. To address this, we have now included **ground-truth transcriptions** for the relevant OCR tasks. For future work, we also plan to release **image-text pairs** to further isolate OCR from cultural knowledge and help models better learn to handle OCR-specific tasks. If time allows, we will conduct additional experiments to test **image + text** interaction more thoroughly.
>
> ---
>
> ### Summary of Manuscript Changes:
>
> - **Section 3.3** — Explicitly clarifies the manual collection cost and efforts for data augmentation.
> - **Section 5.4 & Table 7** — Provides further justification for the observed transfer improvement on MMMU after fine-tuning on TaiwanVQA.
>
> We hope these clarifications adequately address the reviewer's concerns and improve the understanding of TaiwanVQA's strengths and limitations.

---

> > ### Author Response · Authors · 2025-08-07
> > **Response to Weakness 3: "Although the framework is pitched as 'generalizable,' all experiments are Taiwan-specific; transfer to other low-resource cultures is unproven. Later on, QA tasks emphasise picking from four options rather than producing free-form answers, so they mainly test recognition + shallow elimination, not open-ended cultural reasoning."**
> >
> > Dear Reviewer zmFN,
> >
> > While our original submission focused on MCQ format for structured evaluation, **we have now conducted comprehensive Open-QA (free-form) experiments** that validate your concern and provide deeper insights.
> >
> > ## **New Experimental Results: Open-QA Evaluation**
> >
> > > Note: We have currently tested the most popular/representative models from our MCQ evaluation. We will complete the full model suite (all models from our MCQ experiments) for the camera-ready version.
> > >
> >
> > | Model | Overall | w/ OCR | w/o OCR | All (Recognition) | Basic | External | Complex | All (Reasoning) |
> > | --- | --- | --- | --- | --- | --- | --- | --- | --- |
> > | **GPT-4o** | 67.40 | 86.05 | 73.02 | 77.50 | 67.35 | 53.64 | 57.32 | 57.30 |
> > | **GPT-o3** | 68.95 | 84.88 | 73.93 | 77.70 | 67.35 | 57.95 | 57.32 | 60.20 |
> > | **Gemini-2.5-flash** | 66.80 | 86.92 | 70.88 | 76.40 | 71.43 | 52.01 | 57.32 | 57.20 |
> > | **Gemini-2.5-pro** | 71.90 | 89.53 | 74.24 | 79.50 | 74.29 | 60.62 | 64.63 | 64.30 |
> > | **Qwen2.5-VL-32B-Instruct** | 55.85 | 83.14 | 58.23 | 66.80 | 61.63 | 37.59 | 54.88 | 44.90 |
> > | **Qwen2.5-VL-72B-Instruct** | 58.35 | 84.01 | 62.65 | 70.00 | 66.12 | 39.23 | 50.00 | 46.70 |
> > | **InternVL3-38B-Instruct** | 51.65 | 80.23 | 57.62 | 65.40 | 57.55 | 30.91 | 36.59 | 37.90 |
> > | **InternVL3-78B-Instruct** | 53.05 | 81.40 | 57.62 | 65.80 | 61.22 | 32.40 | 42.68 | 40.30 |
> > | **GLM-4.1V-9B-Thinking** | 52.80 | 82.56 | 57.93 | 66.40 | 57.55 | 31.80 | 45.12 | 39.20 |
> > | **MiMo-VL-7B-RL** | 52.80 | 77.91 | 55.95 | 63.50 | 61.22 | 34.32 | 48.78 | 42.10 |
> >
> > ## Key Findings
> >
> > The reviewer's insight about "shallow elimination" vs "open-ended cultural reasoning" is confirmed:
> >
> > 1. **Open-QA is substantially harder**: All models drop 16-30% in overall performance, with reasoning tasks showing even steeper declines
> > 2. **Challenge shifts from visual to cultural**: In MCQ, "Complex" visual reasoning is hardest; in Open-QA, "External" cultural knowledge becomes the bottleneck—models cannot generate Taiwan-specific facts without answer scaffolding
> > 3. **True generative reasoning exposed**: The format change reveals which models truly internalized cultural knowledge (Gemini) vs those relying on elimination strategies (InternVL)
> >
> > ## Camera-Ready Updates
> >
> > - Complete Open-QA evaluation for all models
> > - New section analyzing MCQ vs Open-QA to address "shallow elimination" concerns
> > - Release Open-QA evaluation scripts to enable comprehensive cultural reasoning assessment
> >
> > We believe these additions address the reviewer's valid concerns about testing depth and demonstrate that TaiwanVQA can evaluate both recognition (via MCQ) and true generative cultural reasoning (via Open-QA).

---

### Author Response · Authors · 2025-08-09

Dear Reviewers, Area Chair, and Program Chairs,

We sincerely thank all reviewers for their valuable feedback. Following suggestions from Reviewers zmFN, mnb4, and Phav, we conducted comprehensive **Open-QA experiments** (20-30% performance drop vs MCQ) and expanded evaluation to **20 models**, enriching our contribution.

## Our Core Contribution

TaiwanVQA offers **2,736 originally-photographed images** with **100% human-written questions**, providing:

- First VQA benchmark in Traditional Chinese grounded in Taiwan’s cultural context
- Zero contamination with existing training corpora
- Framework transferable to other underrepresented cultures
- Evidence that even 78B-parameter models struggle with cultural reasoning

## On Regional Specificity

We respectfully maintain that regional cultural understanding is valuable for AI research. Just as NeurIPS has welcomed work on specific languages and regions, Taiwan's unique visual culture—shaped by Indigenous heritage and distinct historical development—offers important insights for building culturally-aware AI systems.

Visual elements like street scenes, signage systems, and cultural symbols cannot be captured through translation alone. They require dedicated study to understand how AI systems perceive and reason about different cultural contexts.

## Moving Forward

We believe TaiwanVQA contributes to the important goal of developing AI that serves diverse communities effectively. The strong support from three reviewers and our extensive experimental validation demonstrate the work's scientific merit.

Thank you for your thoughtful consideration.

Best regards,

The TaiwanVQA Team

---

### Decision · Program_Chairs · 2025-09-18

**Decision:**

Accept (poster)

**Comment:**

The authors present a dataset with 2736 human written questions pertaining to Taiwanese cultural reasoning. After reading the paper, I think it is a great contribution and an interesting benchmark for a use-case that is emerging quite naturally for vision-language models deployed at scale, complementing existing work on cultural visual question-answering. The authors have addressed most of my (and raised by other reviewers') concerns, specifically on the MCQ format vs open-ended format, and around RAI (addressing biases, PII etc. with appropriate declaration. I think the biggest positive for the benchmark is the sourcing of images done entirely by Taiwanese residents, making it valuable both as OOD data for models pretrained on web corpora, as well as representing in-domain queries for VLM users (only limitation being that it was collected via only 9 participants, but the diversity and collection guidelines ensure that they encompass a spectrum of cultural contexts). I recommend a poster acceptance.

When it comes specifically to "domain being narrow", I am not sure I agree -- fine-grained visual categorization precisely tackles tail knowledge concepts, and I think this contribution lies squarely in the "fine-grained visual question answering" domain.